# Oestrogen receptor α AF-1 and AF-2 domains have cell population-specific functions in the mammary epithelium

Stéphanie Cagnet[1], Dalya Ataca [1], George Sflomos[1], Patrick Aouad[1], Sonia Schuepbach-Mallepell[2], Henry Hugues[3], Andrée Krust[4], Ayyakkannu Ayyanan[1], Valentina Scabia[1] & Cathrin Brisken [1]

Oestrogen receptor α (ERα) is a transcription factor with ligand-independent and ligand-dependent activation functions (AF)-1 and -2. Oestrogens control postnatal mammary gland development acting on a subset of mammary epithelial cells (MECs), termed sensor cells, which are ERα-positive by immunohistochemistry (IHC) and secrete paracrine factors, which stimulate ERα-negative responder cells. Here we show that deletion of AF-1 or AF-2 blocks pubertal ductal growth and subsequent development because both are required for expression of essential paracrine mediators. Thirty percent of the luminal cells are ERα-negative by IHC but express *Esr1* transcripts. This low level ERα expression through AF-2 is essential for cell expansion during puberty and growth-inhibitory during pregnancy. Cell-intrinsic ERα is not required for cell proliferation nor for secretory differentiation but controls transcript levels of cell motility and cell adhesion genes and a stem cell and epithelial mesenchymal transition (EMT) signature identifying ERα as a key regulator of mammary epithelial cell plasticity.

---

[1] Swiss Institute for Experimental Cancer Research, School of Life Sciences, Ecole Polytechnique Fédérale de Lausanne, CH-1015 Lausanne, Switzerland. [2] Department of Biochemistry, University of Lausanne, CH-1066 Epalinges, Switzerland. [3] Centre Hospitalier Universitaire Vaudois, Department of Laboratory Medecine, University Hospital of Lausanne, CH-1011 Lausanne, Switzerland. [4] Institut de Génétique et de Biologie Moléculaire et Cellulaire (CNRS UMR7104; INSERM U596; ULP, Collège de France) and Institut Clinique de la Souris, Illkirch, Strasbourg, France. These authors contributed equally: Stéphanie Cagnet, Dalya Ataca, George Sflomos. Correspondence and requests for materials should be addressed to C.B. (email: cathrin.brisken@epfl.ch)

Oestrogens, 17β-estradiol (E2) and its metabolites, are pivotal for the development and the physiology of the breast and impinge on breast carcinogenesis. The oestrogen receptor α (ERα) is expressed in 40% of the luminal cells that make up the inner layer of the mammary epithelium surrounded by basal/myoepithelial cells[1]. Oestrogens drive pubertal development in the mouse mammary gland and induce expression of the progesterone receptor (PgR), activation of which drives cell proliferation during subsequent oestrous cycling and pregnancy. Both hormones rely on paracrine factors to activate stem cells and induce proliferation of other mammary epithelial cells (MECs)[2].

The ERα belongs to the nuclear receptor family and is composed of six modular domains, namely, A to F[3]. Ligand-independent and ligand-dependent activation functions, AF-1 and AF-2 map to the A/B and E domains, respectively[4,5]. Ligand-independent signalling results from phosphorylation of different serine residues in AF-1 by for instance MAPK[6], GSK-3[7] or cyclinA/cdk2[8]. Upon activation, the receptor dimerises and translocates to the nucleus where it interacts either directly with the DNA via specific DNA sequences known as the oestrogen response elements, or indirectly via DNA-binding proteins like AP-1[9]. Full ligand-dependent transcriptional activity relies on synergistic activities of AF-1 and AF-2[5]. A small fraction of the ERα is found at the plasma membrane; it elicits rapid, non-genomic responses, which modulate multiple signalling pathways and create cross-talk between membrane and nuclear ERα[10].

More than 70% of all breast cancers express the ERα and this is exploited therapeutically. The most widely used agent, tamoxifen, antagonises AF-2[11] and agonises AF-1[12], and is used in primary and secondary breast cancer prevention. Most insights into the molecular mechanisms underlying ERα signalling stem from in vitro studies with ERα-positive (ERα+) breast cancer cell lines, in particular MCF-7 cells which express very high levels of the receptor and are exquisitely sensitive to E2. How ER signalling occurs in vivo in normal and cancerous tissue is poorly understood. To dissect the different aspects of ERα signalling in vivo, mice lacking specifically the AF-1 domain (AF-1⁰)[13] or the AF-2 domain (ERαAF-2⁰)[14] were generated and compared to mice lacking the entire ERα (ERα⁻/⁻); all three strains are viable but have impaired reproductive functions, and distinct organ defects[13–15]. The role of the AF-1 and the AF-2 domains in the mammary epithelium was not analysed. Using ERα⁻/⁻ mice, we have previously shown that ERα is required for ductal elongation in the mammary epithelium[16].

Here, we explore the role of AF-1 and AF-2 vs. intact ERα signalling in mammary gland development; we demonstrate differential roles that are dependent on cell type and/or ERα protein levels and uncover important functions of the ERα in apparently ERα-luminal responder cells.

## Results

**Mammary gland development in ERαAF-1⁰ and ERαAF-2⁰ mice.** To assess the impact of germ-line deletion of ERα ligand-dependent, AF-2, vs. ligand-independent, AF-1, genomic actions on mammary gland development, we analysed mammary glands of AF-1⁰ and AF-2⁰ females and their respective WT littermates (Fig. 1a) at critical developmental stages using whole-mount stereomicroscopy (Fig. 1b, Supplementary Figure 1a–d). Before the onset of ovarian function, on postnatal day 21, all females had rudimentary ductal systems (Supplementary Figure 1a) with on average 4.7% fat pad filling in WT and <3% fat pad filling in the ERα mutant littermates (Fig. 1c, Supplementary Figure 1a). In pubertal, that is 4- to 7-week-old WT females, rapidly growing ductal tips enlarged to form terminal end buds (TEBs) and ducts

extended beyond the sub-iliac lymph node to fill 61% of the fat pad (Fig. 1b, c). In AF-1⁰ and AF-2⁰ littermates, no TEBs were found and fat pad filling remained <3% (Fig. 1b, c). In adult, 8- to 12-week-old WT females, fat pads were filled up to 80%, in their AF-1⁰ and AF-2⁰ littermates to 5.1% and 3% only, respectively (Fig. 1c, Supplementary Figure 1b). In older WT females, which have been exposed to repeated oestrous cycle related peaks of E2 and progesterone, side branching occurred (Supplementary Figure 1c, d; Fig. 1c) whereas the block of ductal growth persisted in AF-2⁰ females (Supplementary Figure 1b, c) as observed in ERα⁻/⁻ females[16]. In older AF-1⁰ females, few ducts occasionally extended beyond the lymph node (Supplementary Figure 1c, d, Fig. 1c). Mutant ducts were atrophic with decreased ductal diameters (Supplementary Figure 1c, d) but structurally intact with luminal and basal cell layers as revealed by histological analysis (Supplementary Figure 1e). Immunohistochemistry (IHC) revealed expression of ERα protein in AF-1⁰ and AF-2⁰ mammary epithelia at levels comparable to WT controls (Fig. 1d), as reported for their uteri[13,14]. This excluded the possibility that the mutant ERα proteins were unstable and their expression in MECs was reduced or lost. Thus, the phenotypes reflect the specific deletions of AF-1 or AF-2 domain and show that both are required for ERα function during ductal elongation.

**Endocrine disturbances in AF-1⁰ and AF-2⁰ mice.** Adult AF-1⁰, AF-2⁰ and ERα⁻/⁻ female mice were reported to have increased serum levels of E2, luteinizing hormone and testosterone as measured by radioimmunoassay[17]. Our finding that mammary gland development was already affected before puberty begged the question whether endocrine disturbances may occur earlier. As plasma steroid hormone levels are very low prior to puberty, we used liquid chromatography tandem–mass spectrometry (LC–MS). We confirmed that both E2 and testosterone levels increased significantly in all three mutants in adulthood. E2 was detected at levels of 12 pg/ml in plasma of WT females and increased almost 2-fold in AF-1⁰ and threefold in AF-2⁰ and ERα⁻/⁻ females. Testosterone levels raised from on average 0.1 ng/ml in plasma of WT females to 0.7 ng/ml plasma levels in AF-1⁰, 1.5 ng/ml in AF-2⁰ and 1.4 ng/ml in ERα⁻/⁻ females (Fig. 2a). Plasma progesterone levels varied in WT females due to oestrous cycling not observed in the mutants in which ovarian cycles are not established. All three mutants had decreased 11-deoxycorticosterone levels while 17-OH-progesterone levels were increased in ERα⁻/⁻ females (Fig. 2a).

Progesterone levels increased specifically in peripubertal AF-2⁰ females whereas androstenedione levels increased in peripubertal ERα⁻/⁻ and pubertal AF-2⁰ females (Fig. 2a). Testosterone was significantly increased in peripubertal AF-1⁰ females. Hence, complex alterations of the endocrine milieu occurred already before and during puberty.

An overall analysis of the average fold-changes in steroid levels in ERα mutants compared to WT and heterozygous littermates in relation to their biosynthetic pathways revealed comparable decrease of plasma levels of the progesterone derivatives, 11-deoxycorticosterone and corticosterone, in all mutants in adulthood (Fig. 2b). The levels of E2 and its precursors 17OH-progesterone, androstenedione, and testosterone, were close to WT levels in AF-1⁰ mice but deviated similarly in ERα⁻/⁻ and AF-2⁰ females are in line with negative feedback loops of the E2 biosynthetic pathway being regulated by ligand-dependent ER activity.

**Mammary epithelial-intrinsic role of AF-1 and AF-2 domains.** The systemic effects of the germ-line mutations and the resulting complex endocrine abnormalities confounded the interpretation of the mammary gland phenotype. To assess the MEC intrinsic

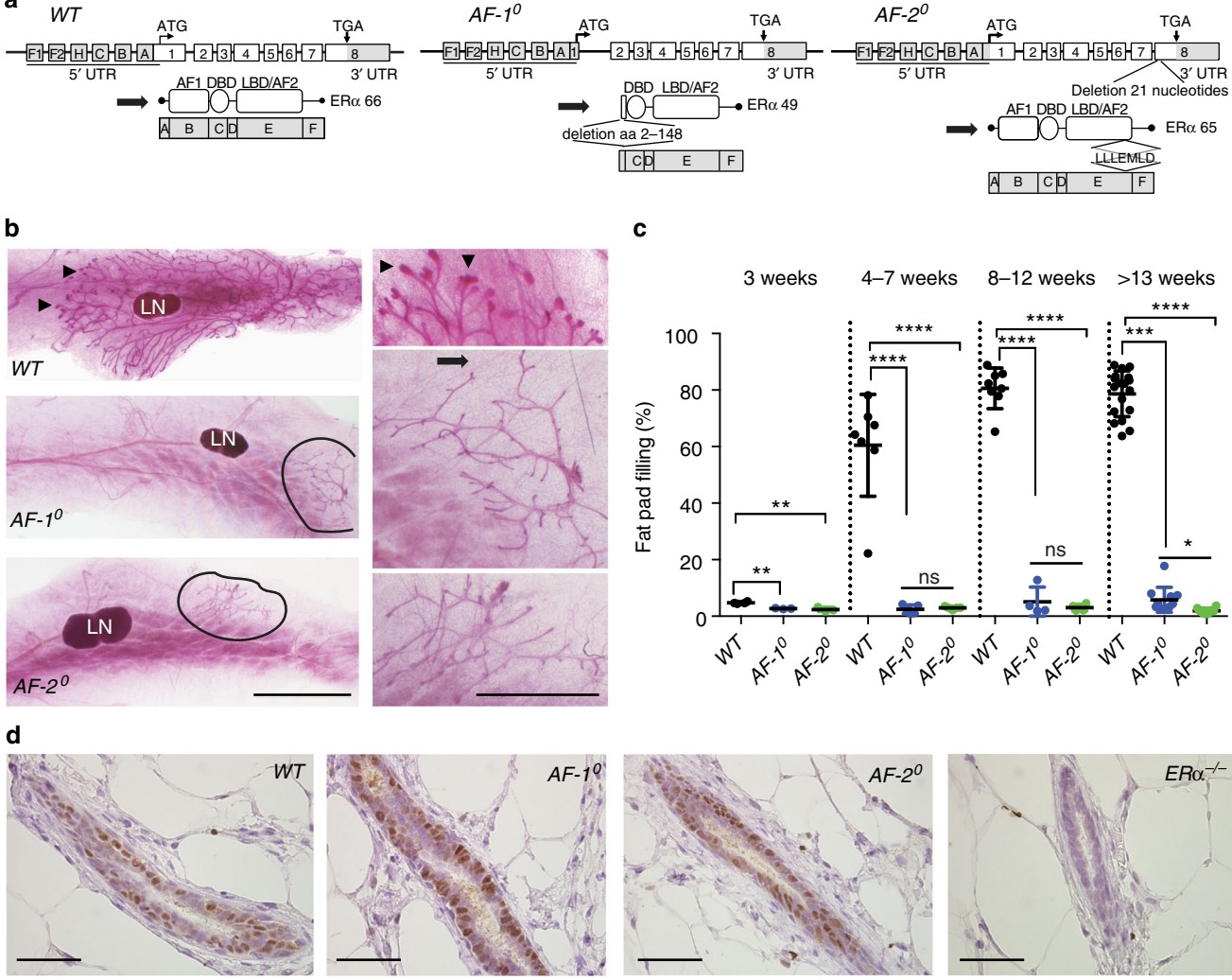

**Fig. 1** Mammary gland phenotype of AF-1⁰ and AF-2⁰ mice. **a** Schematic representation of the mutant genomic loci and the ERα proteins expressed in WT mice, AF-1⁰ mice with deletion of amino acids 2–148, and AF-2⁰ mice with deletion of amino acids 543–549. Only the main protein initiated by the translational initiation codon in exon 1 on amino acid 1 (ATG) is shown for each genotype. The less abundantly expressed protein is initiated in exon 2 on amino acid 178. **b** Whole-mount stereomicrographs of inguinal mammary glands from 7-week-old WT, AF-1⁰ and AF-2⁰ females. Arrowheads indicate TEBs. Black lines mark the borders of the ductal outgrowth. Scale bars: 5 mm (left), 2 mm (right). LN sub-iliac lymph node. **c** Dot plot showing extent of fat pad area filled by the engrafted WT, AF-1⁰ and AF-2⁰ epithelia at different developmental stages (n = 3–18). Shown are means ± SEM; unpaired, two-tailed, Student's t test, *p < 0.05, **p < 0.01, ***p < 0.001, ****p < 0.0001, n.s. not significant. **d** ERα IHC of mammary glands from 3-week-old WT, AF-1⁰, AF-2⁰ and ERα⁻/⁻ mice. Representative pictures of glands analysed from three females of each genotype are shown. Scale bar: 100 μm

requirement of each domain, we grafted fragments of ducts from AF-1⁰.GFP⁺ or AF-2⁰.GFP⁺ and WT.GFP⁺ littermates into contralateral inguinal mammary glands of 3-week-old WT hosts surgically divested of endogenous epithelium[18]. Fluorescence stereomicroscopy of the engrafted glands 10 weeks after surgery showed that grafted WT epithelia filled the fat pads (Fig. 3a, e) but the AF-1⁰.GFP⁺ fragments failed to grow (Fig. 3b, e). During pregnancy, WT.GFP⁺ grafts formed complex ductal trees with alveoli (Fig. 3c, e). Only 1 out of 5 AF-1⁰.GFP⁺ grafts showed some ductal growth with 15% fat pad filling; alveoli were absent (Fig. 3d, e). When the WT grafts developed normally (Fig. 3f, h, j), the contralateral AF-2⁰.GFP⁺ grafts grew neither in nulliparous nor pregnant hosts (Fig. 3g, i, j). Thus, both AF-1 and AF-2 are required in MECs for ductal outgrowth[19,20], side branching, and alveologenesis. AF-1 abrogation is compatible with limited ductal elongation during pregnancy, whereas AF-2⁰ MECs fully recapitulate the ERα⁻/⁻ MEC phenotype with a complete developmental block[16].

Steroid hormones induce mammary gland development largely through paracrine signalling[2]. Areg is an essential mediator of E2-induced cell proliferation during puberty[19] and Wnt4 an important activator of stem cells, which can be induced by E2 in pubertal mammary glands[21]. In mammary glands from pubertal age AF-1⁰ and AF-2⁰ mice, Areg and Wnt4 transcript levels were as low as in their ERα⁻/⁻ counterparts, i.e.,<1 or 10%, respectively, as compared to WT (Fig. 3k). Thus, transcription of Areg and Wnt4 requires both ERα AF-1 and AF-2, providing a mechanism for the growth defect in the mutant epithelia. Transcript levels of other ERα targets, Pgr1[22] and Prlr[23], were similarly decreased (Fig. 3k). However, specifically Pgr1 levels were higher in AF-1⁰ than in AF-2⁰ and ERα⁻/⁻ mammary glands (Fig. 3k). The PgR protein was readily detected by IHC in luminal cells of WT and AF-1⁰, but not of AF-2⁰ or ERα⁻/⁻ females (Fig. 3l) indicating that PgR expression is largely AF-2 dependent and somewhat AF-1 independent.

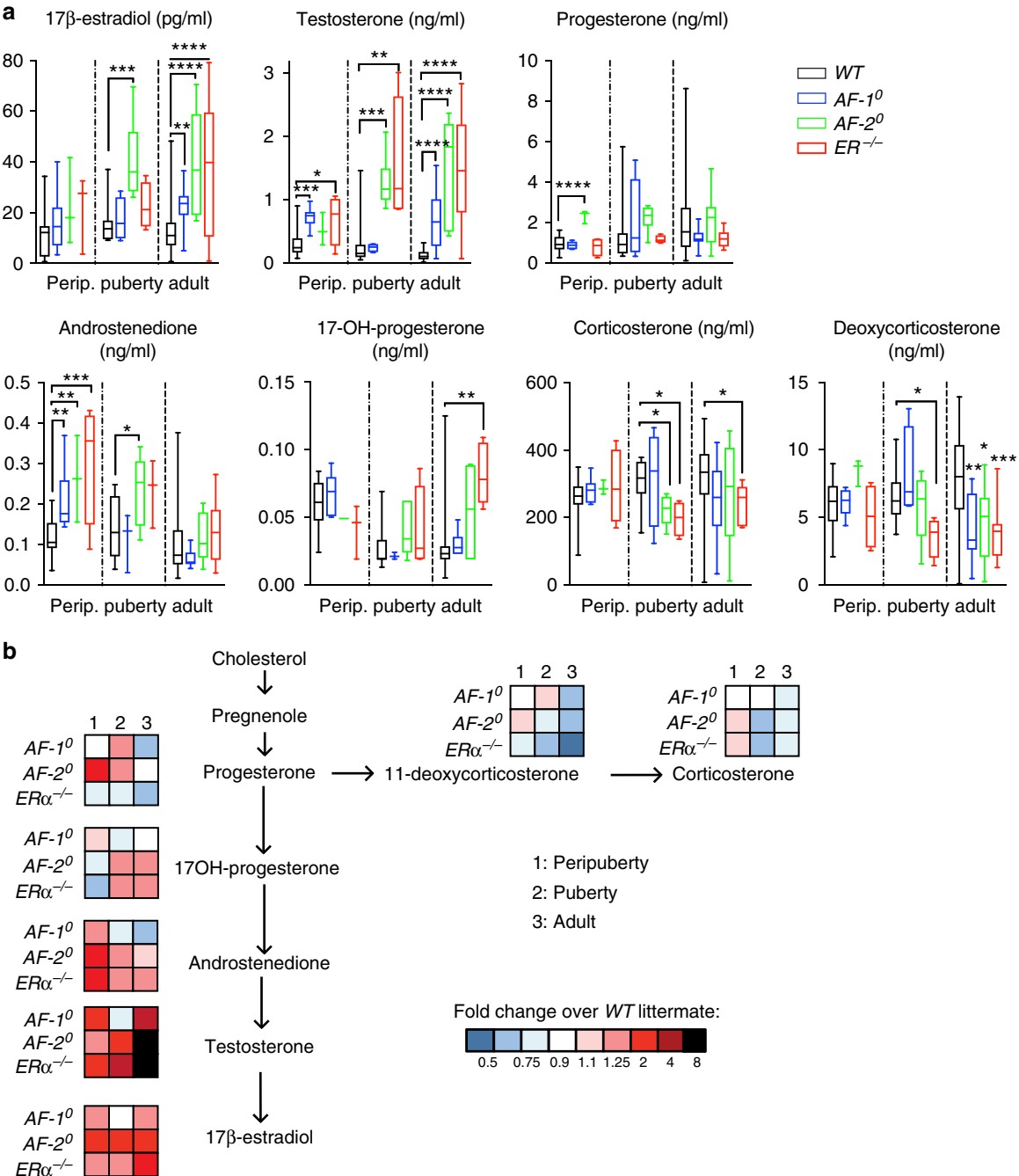

**Fig. 2** Steroid hormone levels in *ERα AF-1⁰*, *ERα AF-2⁰* and *ERα⁻ᐟ⁻* mice. **a** Box plots showing plasma levels of 17-β-estradiol, testosterone, progesterone, androstenedione, 17-OH-progesterone, corticosterone and deoxycorticosterone determined by LC/MS measured in peripubertal (3-week old), pubertal (4- to 7-week old) and adult (>8-week old) *WT*, *AF-1⁰*, *AF-2⁰* and *ERα⁻ᐟ⁻* females (*n* = 3–25). Horizontal lines outside the boxes depict minimum and maximum values, upper and lower borders of the boxes represent lower and upper quartiles, and the line inside the box identifies the median. Unpaired, Mann–Whitney test. **b** Mean serum levels of the different steroids (shown in **a**) were plotted relative to average levels in *WT* littermates in the context of major biosynthetic pathways. Colour code represents fold changes over levels in the *WT* littermates

We hypothesised that the residual PgR expression in *AF-1⁰* epithelium may account for the partial outgrowth in this mutant observed in the older females and during pregnancy. To test whether prolonged PgR signalling is sufficient to stimulate ductal growth, we exposed mice bearing *AF-1⁰.GFP⁺* and *WT.GFP⁺* epithelial grafts to pregnancy levels of progesterone by implanting slow-release pellets subcutaneously. Serum progesterone levels at sacrifice corresponded to those observed during pregnancy in control animals (Supplementary Figure 2a). While the

contralateral *WT.GFP⁺* grafts showed increased side branching (Supplementary Figure 2b) confirming successful delivery of progesterone, the *AF-1⁰.GFP⁺* epithelial grafts remained rudimentary (Supplementary Figure 2c) indicating that activation of PgR signalling is not sufficient to stimulate ductal elongation in *AF-1⁰* epithelial grafts. These observations support a model where the function of ER+ sensor cells relies on both AF-1 and AF-2 for the transcription of *Areg* and *Wnt4* and potentially other factors and/or cell-intrinsic functions required for ductal elongation.

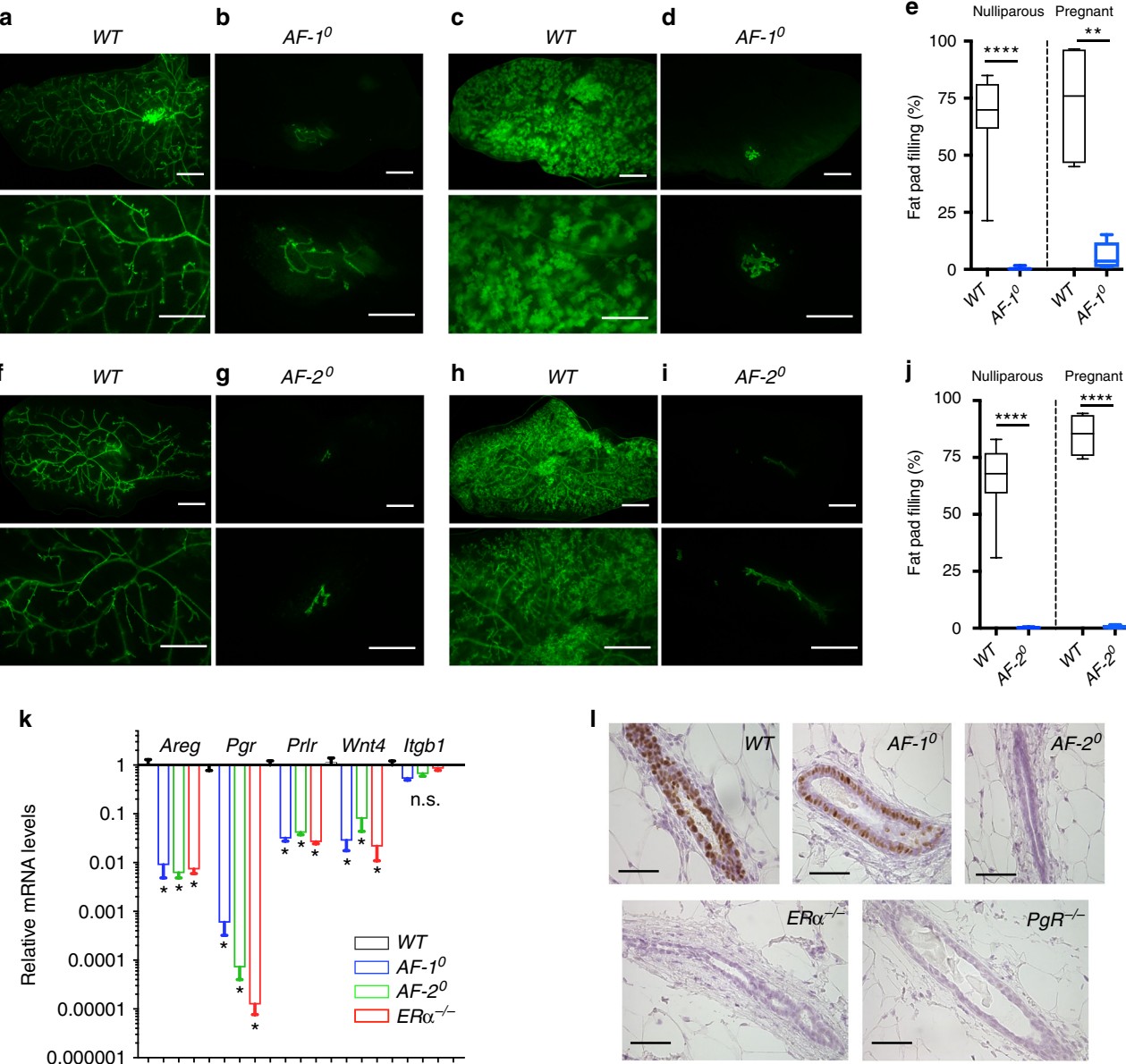

**Fig. 3** Mammary epithelial-intrinsic role of the ERα AF-1 and AF-2 domains. **a–d** Fluorescence stereomicrographs of contralateral inguinal mammary fat pads engrafted with mammary epithelium from *AF-1⁰.GFP⁺* or *WT.GFP⁺* littermates. Nulliparous (**a**, **b**) and day 16–18 pregnant (**c**, **d**) recipients are shown. Scale bars; 5 mm (top), 2 mm (bottom). **e** Box plot showing extent of fat pad filling by the engrafted epithelia in virgin (*n* = 18) and pregnant (*n* = 5) recipients. **f–j** Fluorescence stereomicroscopy of contralateral inguinal mammary fat pads engrafted with mammary epithelium from *AF-2⁰.GFP⁺* or *WT.GFP⁺* littermates. Nulliparous (**f**, **g**) and (P16–18) pregnant (**h**, **i**) recipients are shown. Scale bars; 5 mm (top) 2 mm (bottom). **j** Box plot showing extent of fat pad filling by the engrafted epithelia in virgin (*n* = 10) and pregnant (*n* = 4) recipients. For both box plots, horizontal lines outside the boxes depict minimum and maximum values, upper and lower borders of the box represent lower and upper quartiles and the line inside the box identifies the median. **k** Bar plot showing relative transcript levels of the ERα target genes *Areg*, *Pgr1*, *Prlr* and *Wnt4*, and a control gene, *Itgb1*, normalised to *36b4* and *Hprt* in mammary glands from peripubertal *WT*, *AF-1⁰*, *AF-2⁰* and *ERα⁻/⁻* females. Data are shown as means ± SEM of three independent experiments. Paired two-tailed Student's *t* test, *$p < 0.05$, **$p < 0.01$, ***$p < 0.001$, ****$p < 0.0001$, n.s. not significant. **l** PgR IHC of mammary glands from 3-week-old *WT*, *AF-1⁰*, and *AF-2⁰*, *ERα⁻/⁻* and *PR⁻/⁻* mice. Representative pictures of glands analysed from three females of each genotype are shown. Scale bar; 100 μm

**The role of ERα in responder cells in ductal outgrowth.** We previously reported that when *ERα⁻/⁻ROSA26⁺* MECs were mingled with excess *WT* MECs and this mixture was then grafted to cleared mammary fat pads, *ERα⁻/⁻* MECs contributed to the chimeric outgrowth[16] and gave rise to the model of sensor and responder cells[24,25]. To compare the ability of *AF-1⁰*, *AF-2⁰* and *ERα⁻/⁻* MECs, none of which were able to grow out on their own, to contribute to mammary gland development in the context of excess *WT* MECs that release paracrine and potentially other signals, we co-injected either *ERαWT.GFP⁺* or *ERα* mutant.

*GFP⁺* MECs mixed with *WT.DsRed⁺* MECs in a 1:10 ratio (Fig. 4a) into contralateral glands. The resulting *WT.GFP⁺:WT.DsRed⁺* chimeric glands appeared to 4% GFP⁺, almost 10 times as many seemed DsRed+ (37%), while 59% scored double positive by fluorescence stereomicroscopy (Fig. 4b, c). In the *ERα⁻/⁻.GFP⁺:WT.DsRed⁺* chimeras, only 23% were double positive whereas 77% appeared DsRed+ (Fig. 4b, c). This indicates that *ERα⁻/⁻.GFP⁺* MECs are able to proliferate when mixed with *WT.DsRed⁺* MECs, but are less efficient than *ERαWT.GFP⁺* MECs in contributing to ductal outgrowth. This demonstrates

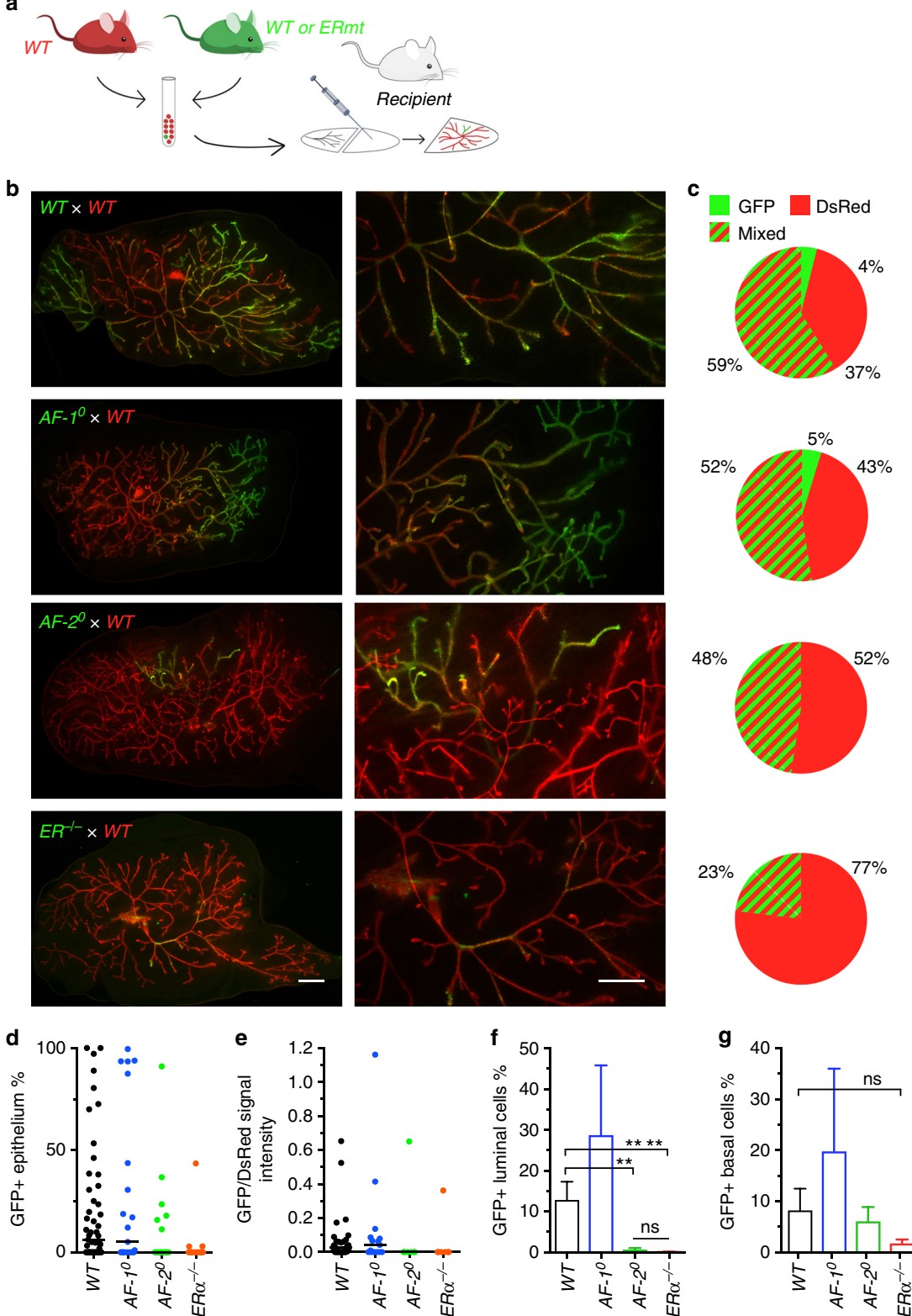

that ERα is required not only in the ERα+ sensor cells but also in the apparently ERα- responder MECs.

To assess whether ERα function in responder cells requires AF-1 and AF-2, we generated chimeras with $AF-1^0$ or $AF-2^0$ cells. $AF-1^0.GFP^+:WT.DsRed^+$ chimeras were comparable to the $WT$ chimeras (Fig. 4b, c). $AF-2^0.GFP^+:WT.DsRed^+$ chimeras were more similar to $ER\alpha^{-/-}.GFP^+:WT.DsRed^+$ chimeras with 52%

appearing DsRed+ only and 48% double positive (Fig. 4b, c). Thus, ERα function in the responder cells is AF-1 independent but AF-2 dependent.

We approximatively evaluated the GFP signal (Fig. 3d) and the ratio of GFP to RFP signal intensities (Fig. 4e) in chimeric epithelia by analysis of a 2D-image of the chimeric gland and showed that the data points were similarly distributed for $WT$ and

**Fig. 4** Contribution of $AF-1^0$, $AF-2^0$ and $ER\alpha^{-/-}$ cells to chimeric ducts with *WT* MECs. **a** Scheme of experimental design. After cell dissociation, 10,000 $ER\alpha$ mutant (*ERmt*) or *WT* $GFP^+$ epithelial cells were mixed with 90,000 *WT.DsRed*$^+$ epithelial cells and injected into the cleared mammary fat pad of peripubertal recipient mice. **b** Representative fluorescence stereomicrographs of chimeric epithelia from *WT.GFP*$^+$ or $ER\alpha.mutant.GFP^+$ and *WT.DsRed*$^+$ cells mixed in a 1:10 ratio. Hosts were analysed 10 weeks after engraftment. Scale bars; 1 mm (left) 0.2 mm (right). **c** Pie charts showing the proportion of engrafted mammary glands appearing exclusively DsRed$^+$, exclusively GFP$^+$, or mixed (red and green stripes) based on evaluation at low (7.8×) magnification of fluorescence stereomicrographs. From top to bottom, $n = 51$, 21, 17 and 13. **d** Dot plot showing the percentage of the reconstituted ductal epithelium that is GFP$^+$ in virgin mice based on images at 7.8× magnification, ($n = 13–51$) bars indicate medians. **e** Dot plot showing the ratio of GFP/DsRed signal intensity of the reconstituted ductal epithelia based on images at 7.8× magnification, bars indicate medians ($n = 29$, 15, 7 and 7). **f, g** Bar graphs showing flow cytometric analysis of the percentage of GFP$^+$ cells in the CD24$^{high}$ CD49f$^{low}$ **f** and CD24$^{low}$ CD49f$^{high}$ **g** cell populations of reconstituted chimeric mammary epithelia. From left to right, $n = 20$, 5, 8 and 6. Shown are means ± SEM; Mann–Whitney test, two-tailed, $^{**}p < 0.01$, $^{****}p < 0.0001$, n.s. not significant

$AF-1^0$ chimeras whereas they tended to be lower for $AF-2^0$ and $ER\alpha^{-/-}$ chimeras (Fig. 4d, e). To determine the contribution of GFP+ MECs quantitatively, we analysed chimeric glands by FACS. After lineage+ cell depletion, we discriminated CD24$^{high}$CD49f$^{low}$ luminal and CD24$^{low}$CD49f$^{high}$ basal cells[26]. *WT.GFP*$^+$ cells represented $12.9 \pm 3.5\%$ of the luminal and $8.2 \pm 2.7\%$ of the basal cell population (Fig. 4f, g) reflecting the original 10% GFP$^+$ cells. The deviation from the predicted 10% may reflect biological variation or cell type-specific differences in expression from the chicken β-actin promoter driving GFP[27,28]. $AF-1^0.GFP^+$ MECs contributed similarly to both cell lineages. $AF-2^0.GFP^+$ and $ER\alpha^{-/-}.GFP^+$ cells were significantly less represented with <1% of the luminal cells (Fig. 4f) and 6% and 1.7%, respectively, of the basal cells but the latter failed to reach statistical significance (Fig. 4g).

Thus, while both AF-1 and AF-2 are required for the expression of essential paracrine mediators in sensor cells, the responder cells require ERα but more specifically AF-2; AF-1 is not required in responder cells during ductal elongation.

**The role of AF-1, AF-2 and ERα during pregnancy**. During pregnancy, E2 levels increase from <10 to 100 pg/ml[29] and there is extensive cell proliferation of MECs, which are hormone receptor negative by IHC[30,31]. Fluorescence stereomicroscopy at the end of pregnancy showed that 90% of *WT.GFP*$^+$:*WT.DsRed*$^+$ chimeras were mixed while 10% appeared DsRed+ only (Fig. 5a, b). Of the $AF-1^0.GFP^+$:*WT.DsRed*$^+$ and $AF-2^0.GFP^+$:*WT.DsRed*$^+$ chimeras 43% scored double positive and 57% DsRed+ only whereas of the $ER\alpha^{-/-}.GFP^+$ chimeras 67% scored double positive (Fig. 5a, b). Approximate evaluation of the GFP signal (Fig. 5c) and the ratio of GFP to RFP signal intensity (Fig. 5d) based on a 2D image showed similar distributions among the chimeras. Comparison of the contribution of the different mutant cells to that of *WT* cells before and at the end of pregnancy indicates that loss of ERα function promotes cell expansion during pregnancy largely through AF-2, whereas AF-1 loss may have an inhibitory effect on cell expansion. Together, the observations on the chimeric outgrowths suggests that ERα function in the ERα low cells is biphasic with a growth-stimulatory role during puberty and an inhibitory role during pregnancy both of which appear AF-2 dependent and AF-1 independent.

**Evaluating ERα in vivo function in by intraductal grafting**. The surprising finding that ERα signalling appeared to have a biphasic effect on MECs depending on the developmental stage incited us to look at ERα in vivo function by an alternative approach that bypasses the need of dissociated MECs to establish themselves in the mammary fat pad because this does not happen physiologically during development and could lead to artefacts, which may confound the interpretation of the results. Human breast epithelial cells have difficulties in establishing themselves in the mouse mammary fat pad but when injected into the milk ducts

they insert into the mouse mammary epithelium and proliferate there[32]. We ascertained that the same holds true for murine MECs by injecting *WT.DsRed*$^+$ MECs intraductally into *NOD scid gamma (NSG).GFP*$^+$ females. Five days later, *WT.DsRed*$^+$ MECs were detected in the GFP$^+$ ducts by fluorescence stereomicroscopy (Fig. 6a). Double immunofluorescence revealed that most DsRed+ cells were luminal MECs, distinct from the P63+ basal cells (Fig. 6b); about 10% of the DsRed+ cells gave rise to myoepithelial cells as identified by smooth muscle actin co-staining (Fig. 6c).

Next, we injected MECs from *WT.GFP*$^+$ and $ER\alpha^{-/-}.GFP^+$ littermates into contralateral glands and analysed them 5 days later by stereomicroscopy. The fluorescent signal from $ER\alpha^{-/-}.GFP^+$ cells was consistently lower than the signal from the contralateral *WT.GFP*$^+$ cells (Fig. 6d, f). When hosts were mated and their engrafted glands were analysed in late pregnancy, the signals became comparable (Fig. 6e, f) suggesting that $ER\alpha^{-/-}$ MECs caught up with their *WT* counterparts as observed in the context of the chimeric epithelial outgrowths above (Fig. 5a). Quantification of GFP+ cells by FACS revealed that *WT.GFP*$^+$ cells represented 4.4% and $ER\alpha^{-/-}.GFP^+$ cells <0.7% of the dissociated cells in virgin mice whereas at day 14.5–16.5 of pregnancy, the contributions of *WT.GFP*$^+$ and $ER\alpha^{-/-}.GFP^+$ MECs were comparable (Fig. 6g, h).

Having ascertained that the biphasic effect of ERα observed in the context of chimeric outgrowths is reproduced in the intraductal model, we evaluated the role of AF-1 and AF-2. Because of the inter-experimental variability in the cell preparations, we normalised the individual $ER\alpha$ mutant cell numbers to the contralateral *WT* cells. In the nulliparous hosts, all three $ER\alpha$ mutants established themselves less well intraductally than their *WT* counterparts; $AF-1^0$ MECs showed 35%, $AF-2^0$ MECs 83%, and $ER\alpha^{-/-}$ MECs 70% reduction, respectively (Fig. 6i). Towards the end of pregnancy, none of the mutants differed significantly from the *WT* controls (Fig. 6j). Thus, by two distinct experimental approaches a biphasic effect of $ER\alpha^{-/-}$ through $AF-2^0$ in MECs is revealed. AF-1 function which was not required for ductal elongation in the context of chimeric glands did affect the ability of MECs to establish themselves intraductally arguing that it may affect cell–cell interactions required for the insertion.

The finding that as early as 5 days after cell injection the number of *WT.GFP*$^+$ MECs that have inserted into the ducts exceeds that of $AF-2^0$ and $ER\alpha^{-/-}$ MECs by several fold, suggested that MECs require ERα to adhere and insert themselves into the host epithelium.

To assess the proliferative indices of $ER\alpha^{-/-}$ MECs and *WT* MECs during puberty and pregnancy we took different approaches. In one setting, we measured cell proliferation of $ER\alpha^{-/-}.GFP^+.WT.DsRed^+$ and *WT.GFP*$^+.WT.DsRed^+$ chimeras during ductal elongation by Ki67 staining because the number of $ER\alpha^{-/-}$ MECs obtained after intraductal injection is very low in virgin recipients. In the other setting, we injected EdU into pregnant mice, which had been intraductally engrafted with ERα

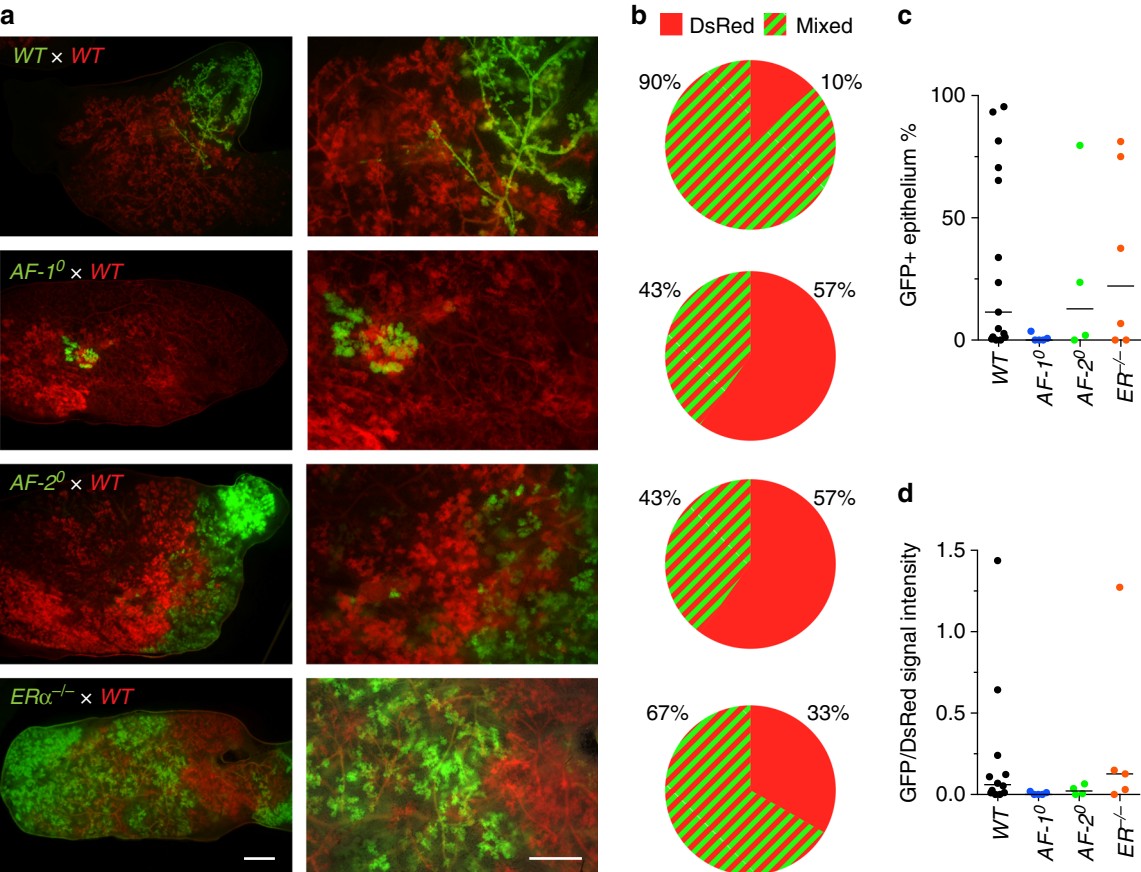

**Fig. 5** Proliferation of *AF-1*[0], *AF-2*[0] and *ERα*[−/−] cells in chimeric epithelia during pregnancy. **a** Representative fluorescence stereomicrographs of chimeric epithelia from *WT.GFP*[+] or *ERα mutant.GFP*[+] and *WT.DsRed*[+] cells mixed in a 1:10 ratio. Hosts were analysed at P16–18. Scale bars; 1 mm. **b** Pie charts showing the number of mammary glands presenting only DsRed[+] epithelial regenerations (red) or mixed [+] and GFP[+] regenerations (red and green stripes) based on evaluation on "low magnification 7.8×" fluorescence stereomicrographs; from top to bottom, *n* = 20, 7, 7 and 6. **c** Dot plot showing percentage of area filled with GFP[+] structures over total area filled in chimeric glands during pregnancy; evaluation at 7.8× magnification; shown are means ± SEM. **d** Dot plot showing ratios of GFP over DsRed signal intensity of different chimeras in pregnant hosts (*n* = 4–16; black line: median)

[−/−].GFP or *AF-2*[0].GFP MECs contralateral of *WT.GFP* MECs and subsequently quantified the GFP and EdU double positive cells. In both scenarios the proliferative indices were comparable between *WT* and *ER* mutant MECs (Fig. 6k, l). Thus, the absence of ER does not impair cell-intrinsic proliferation.

We hypothesised that the results may related to different numbers of stem/progenitor cells present in the mutant mammary epithelia. However, because of the low number of MECs that can be isolated from the *ERα* mutant mammary glands it was not practical to FACS purify any particular subpopulations for in vivo experiments. To assess whether the luminal and basal progenitor cell populations are affected by different *ERα* mutations, we FACS profiled mammary gland cells after lineage[+] cell depletion with CD24 and CD49f[26,33]. While the *WT* profiles were consistent between independent experiments, the *AF-1*[0], *AF-2*[0] and *ERα*[−/−] MECs showed major shifts in both CD24 and CD49f expression, making it impossible to identify the different progenitor populations (Fig. 6m).

**ERα-dependent gene expression in mammary epithelial cells**. To discern the molecular basis of the *ERα*[−/−] MEC phenotypes, and to further test for evidence that stem cell function is impaired by abrogating ERα signalling, *WT.GFP*[+] and *ERα*[−/−].GFP[+] MECs were intraductally injected to contralateral glands, isolated from pregnant hosts by FACS sorting for GFP, and analysed by RNA sequencing. Principal component analysis separated samples by *ERα* genotype (Fig. 7a). A total of 651 genes were differentially expressed, most of which were lower in the *ERα*[−/−]. GFP[+] MECs (>twofold, FDR < 0.05) (Fig. 7b).

Expression levels of genes that mark ERα[high] MECs[34], including *Esr1* itself, were low and no significant differences were found consistent with the scarcity of ERα+ MECs in the pregnant mammary epithelium (Fig. 7c). Milk protein coding genes, were highly expressed in both *WT* and *ERα*[−/−] MECs (Fig. 7d) establishing that ERα is not required for functional differentiation, at least not cell intrinsically. MetaCore terms related to cytoskeleton and cell adhesion were decreased in the *ERα*[−/−] MECs (Fig. 7e) providing potential molecular underpinnings for impaired intraepithelial insertion. MetaCore terms related to immune signalling were increased in *ERα*[−/−] MECs (Fig. 7f). Reactome[35] analysis revealed decreased keratinisation (Fig. 8a) and confirmed increased expression of immunity-related genes in *ERα*[−/−] MECs (Fig. 7g). Moreover, Ephrin and FGFR signalling as well as ECM remodelling were decreased (Fig. 7h); all three have been implicated in pubertal growth[36,37] but had not previously been linked to ERα signalling directly. GO terms and KEGG terms highlighted that the upregulated genes relate to T-cell immunity (Fig. 7i–j).

We noticed basal markers like *Krt5*, *Snai2*, *P63* and *Frizzled7* among the most differentially expressed genes (Fig. 8a). Gene set enrichment analysis for hallmark gene sets[38] showed that an EMT signature was most similar with *p* = 10E28 and compared to

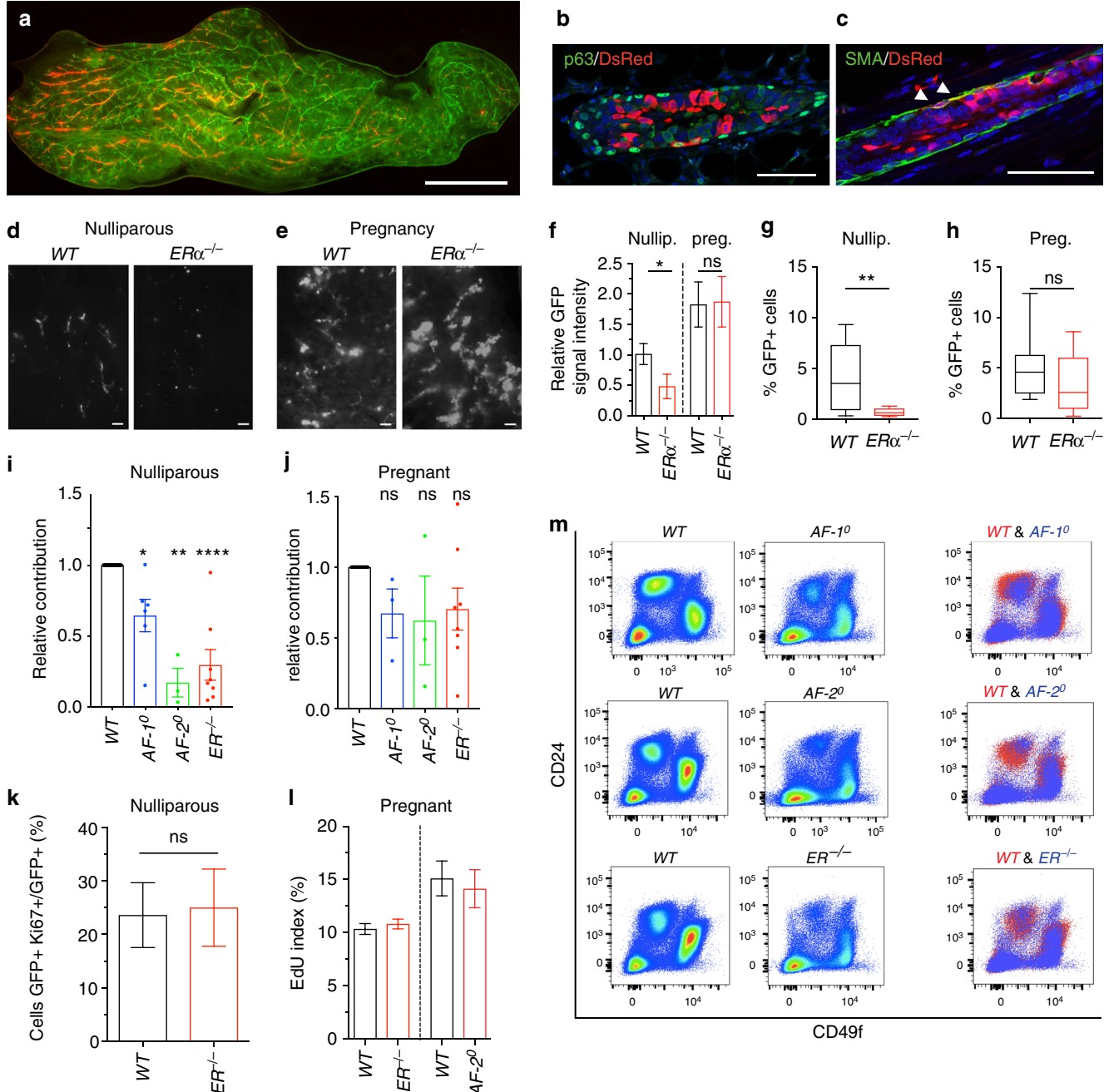

**Fig. 6** Intraductal engraftment of *WT* and *ERα*$^{-/-}$ MECs. **a** Fluorescence stereomicrograph of *NSG.GFP*$^+$ mammary gland 5 days after intraductal injection of *WT.DsRed*$^+$ MECs, representative of nine successfully injected glands. Scale bar; 1 mm. **b, c** Double immunofluorescence with anti-P63 and anti-RFP antibodies **b** or anti-αSma and anti-RFP antibodies **c** counterstained with DAPI. Scale bars; 50 μm. Representative pictures of injected glands from three females. **d, e** Fluorescence stereomicrographs of contralateral mammary glands 5 days after intraductal injection with *WT.GFP*$^+$ or *ERα*$^{-/-}$.*GFP*$^+$ in nulliparous **d** or day 16–18 pregnant host (**e**). Scale bars; 500 μm. **f** Bar plot showing relative GFP signal intensity in contralateral glands injected with *WT. GFP*$^+$ or *ERα*$^{-/-}$.*GFP*$^+$ cells in nulliparous and pregnant recipients, ($n = 3$–6; mean ± SEM). **g, h** Box plots showing percentage of GFP+ cells by FACS in nulliparous **g** and pregnant **h** hosts ($n = 8, 9$). Whiskers depict minimum and maximum values, box borders lower and upper quartiles, line inside identifies the median. Student's paired *t* test, two-tailed. **i, j** Bar plots overlying dot plots showing relative contribution of GFP+ cells by FACS of intraductally engrafted glands from nulliparous (**i**) and pregnant (**j**) hosts, mean ± SEM. **k** Box plot showing percentage of GFP and Ki67 double + over total GFP+ cells from contralateral *WT.GFP*$^+$ or *ERα mutant.GFP*$^+$ with *WT.DsRed*$^+$ MECs chimeras in nulliparous recipients 4 weeks after engraftment ($n = 3$; mean ± SD). **l** Box plot showing the percentage of GFP and EdU double + over total GFP+ cells from contralateral inguinal mammary glands intraductally engrafted with *WT.GFP*$^+$ or *ERα*$^{-/-}$.*GFP*$^+$ (left panel) and *WT.GFP*$^+$ or *AF-2*$^0$.*GFP*$^+$ MECs (right panel) in pregnant recipients P12.5 ($n = 3$; mean ± SD). **m** Representative FACS plots showing CD49f and CD24 expression in Lin$^-$ cells from mammary glands of *AF-1*$^0$, *AF-2*$^0$, *ERα*$^{-/-}$ females and *WT* littermates. For both box plots, whiskers depict minimum and maximum values, borders of the box represent lower and upper quartiles, and line inside the box identifies the median. Paired two-tailed Student's *t* test, $^*p < 0.05$, $^{**}p < 0.01$, $^{***}p < 0.001$, $^{****}p < 0.0001$, n.s. not significant

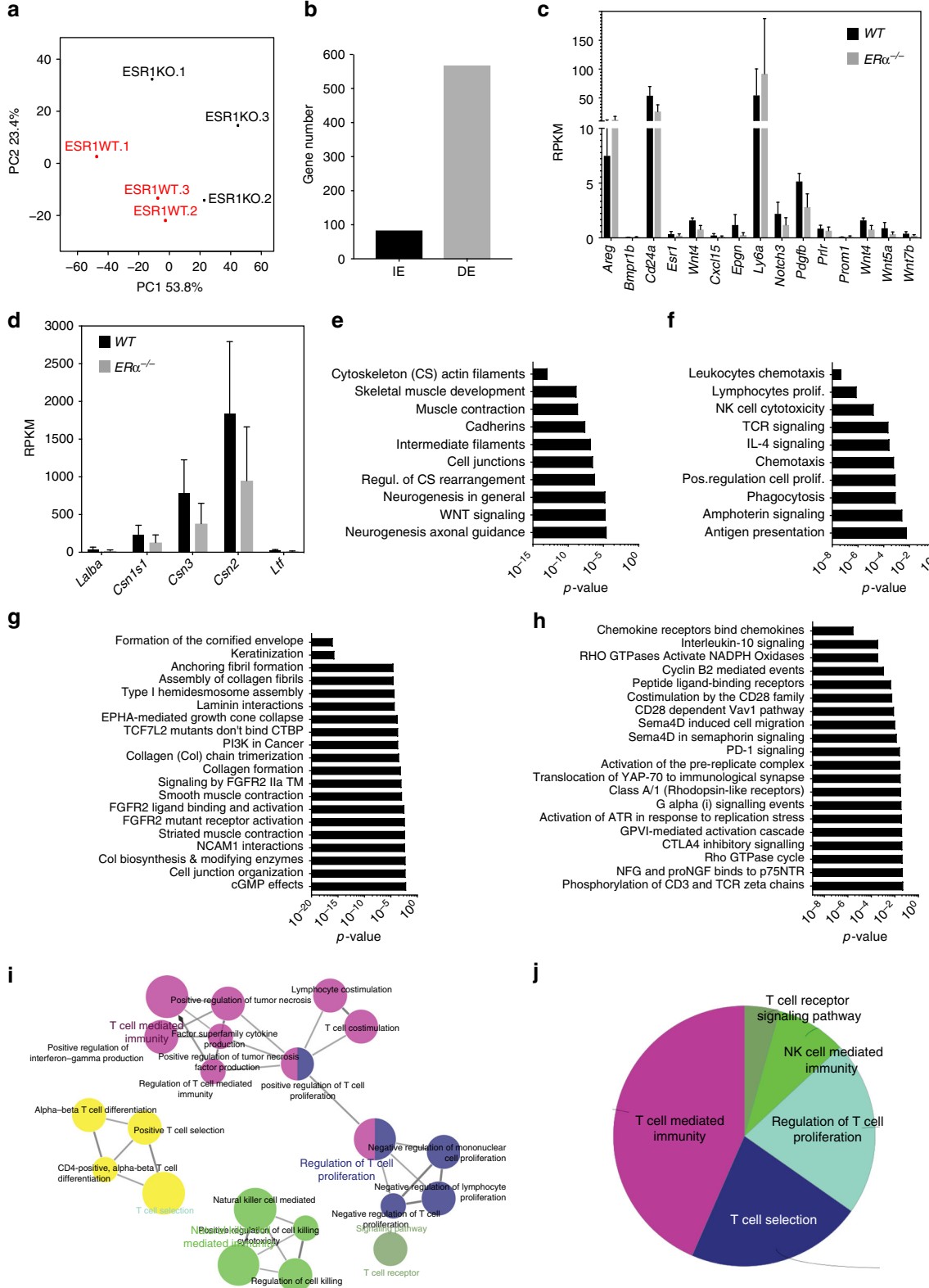

curated gene sets a mammary stem cell signature scored highest with $p = 8E105$ (Fig. 8b, c). Thus, ERα expression is not required for milk gene expression but is important for the transcriptional control of mammary stem/progenitor cell function as well as cellular interactions.

**ERα status of luminal epithelial cells**. Our finding that ERα has an important role in cells that appear ERα negative by IHC begged the question whether IHC may fail to detect functionally relevant ERα protein expression. Hence, we sought to detect *ERα* transcripts in situ in sections adjacent to sections assessed by IF

**Fig. 7** Global gene expression profile of *ERα*−/− and *WT* MECs. **a** Principal component (PC) analysis of RNAseq data showing the importance of the *Esr1* genotype for global gene expression. **b** Bar graph showing the number of genes whose expression increased (IE) or decreased (DE) in *ERα*−/−.*GFP*+ vs. *WT.GFP*+ MECs engrafted intraductally to contralateral glands of *NSG* females and subsequently isolated by FACS-sorting from hosts during pregnancy. **c** Bar plot showing reads per kilobase million (RPKM) of ERα high, sensor cell markers expressed in *ERα*−/−.*GFP* and *WT.GFP*+ MECs that were engrafted intraductally to contralateral glands of *NSG* hosts and subsequently isolated by FACS-sorting from pregnant hosts. **d** Bar plot showing RPKM of milk genes expressed in FACS-sorted *ERα*−/−.*GFP*+ and *WT.GFP*+ MECs grafted intraductally and isolated from pregnant hosts. **e, f** MetaCore analysis of genes showing decreased (**e**) or increased (**f**) expression, in *ERα*−/−.*GFP*+ MECs compared to *WT.GFP*+ MECs. The most significantly enriched terms are listed with *p* value. **g, h** Reactome analysis of genes with decreased **g** or increased **h** expression, in *ERα*−/−.*GFP*+ MECs compared to *WT.GFP*+ MECs. The most significantly enriched terms are listed with *p* value. **i** Visual representation of GO terms with increased expression in *ERα*−/−.*GFP*+ vs. *WT.GFP*+ MECs by Cytoscape. **j** Pie chart showing proportions of KEGG terms among the genes with increased expression in the *ERα*−/−.*GFP*+ vs. *WT.GFP*+ MECs

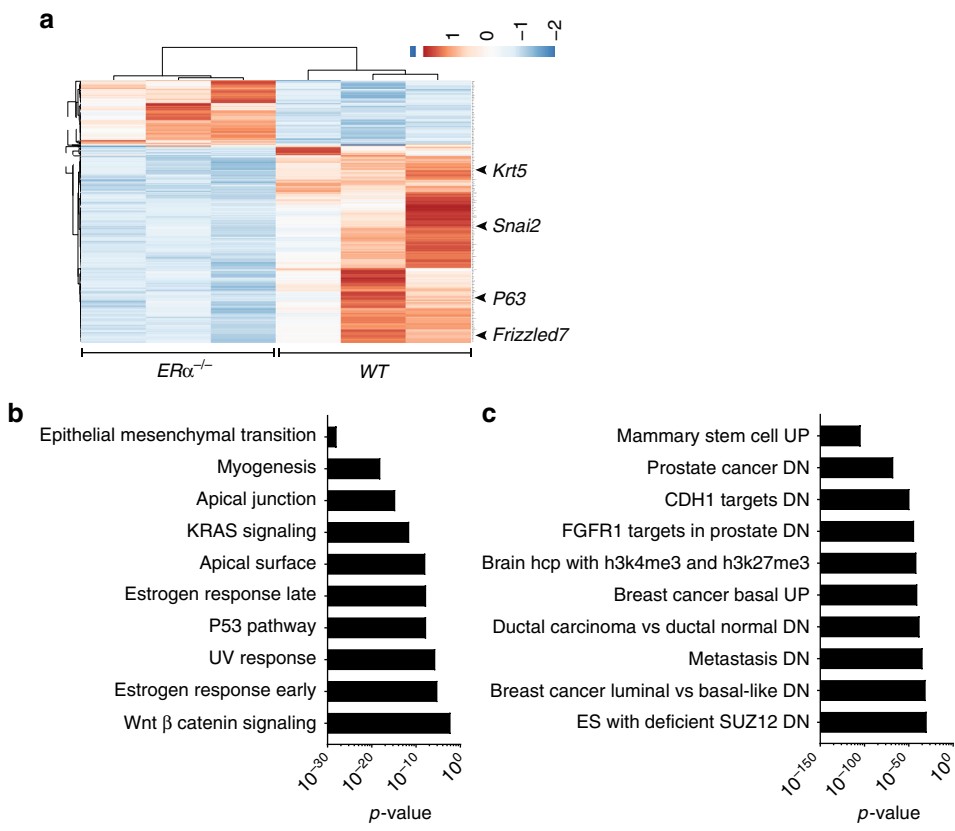

**Fig. 8** ERα-mediated control of gene expression signatures. **a** Heatmap of RNAseq transcriptomic analysis of *WT.GFP*+ and *ERα*−/−.*GFP*+ MECs that grew intraductally in NSG females contralateral glands. Row *Z* scores of genes that were differentially expressed are included. **b, c** Gene set enrichment analysis of genes that were down-regulated, in in *ERα*−/−.*GFP*+ MECs compared to *WT.GFP*+ control cells. By hallmark gene sets (**b**) and curated gene sets (**c**). The most significantly enriched terms with *p* values are listed

for ERα protein expression. While ERα protein was detected in 50% of the luminal cells (Fig. 9a, c), RNAscope detected *ERα* mRNA in 80% of the luminal cells (Fig. 9b, c) indicating transcript expression in luminal cells other than the 50% ERα+ by IHC, the protein may be expressed at levels below the IHC detection limit possibly because of rapid turnover. Semi-quantitative scoring based on the number of dots per cells showed approximately 20% of the luminal cells falling into the negative, the low (1) and medium (2) categories and 40% into the high (≥3) (Fig. 9d), Of note, transcripts of the ERα target *Areg* were only detected in a subset of 15–30% of luminal cells (Fig. 9e), which we assume to be ERα^high sensor cells because they co-express PgR protein, here used as a proxy for ERα since the antibodies for ER do not work with RNAscope (Fig. 9e–g).

Thus, with respect to ERα status, at least three different luminal cell types can be distinguished. The sensor cells, ERα+ by IHC, the responder cells, which are ERα- by IHC yet express detectable

amounts of ERα transcript, and ERα negative responder cells. Sensor cells require both AF-1 and AF-2 whereas the ERα^low responder cells are AF-2-dependent. These three groups may represent three distinct classes of luminal epithelial cells or different zones of a gradient of different ERα expression levels.

## Discussion

The present study of the in vivo role of ERα and its subdomains AF-1 and AF-2 in the mammary epithelium reveals unexpected complexities of this signalling pathway. Contrary to current thinking that there is a dichotomy between ERα+ and ERα- luminal cells, we show that based on ER expression levels, a third luminal cell population, the ERα^low cells, can be distinguished. These cells have readily detectable *Esr1* transcripts but are characterised by low-level ERα protein expression, possibly attributable to high protein turnover. The percentage of luminal cells

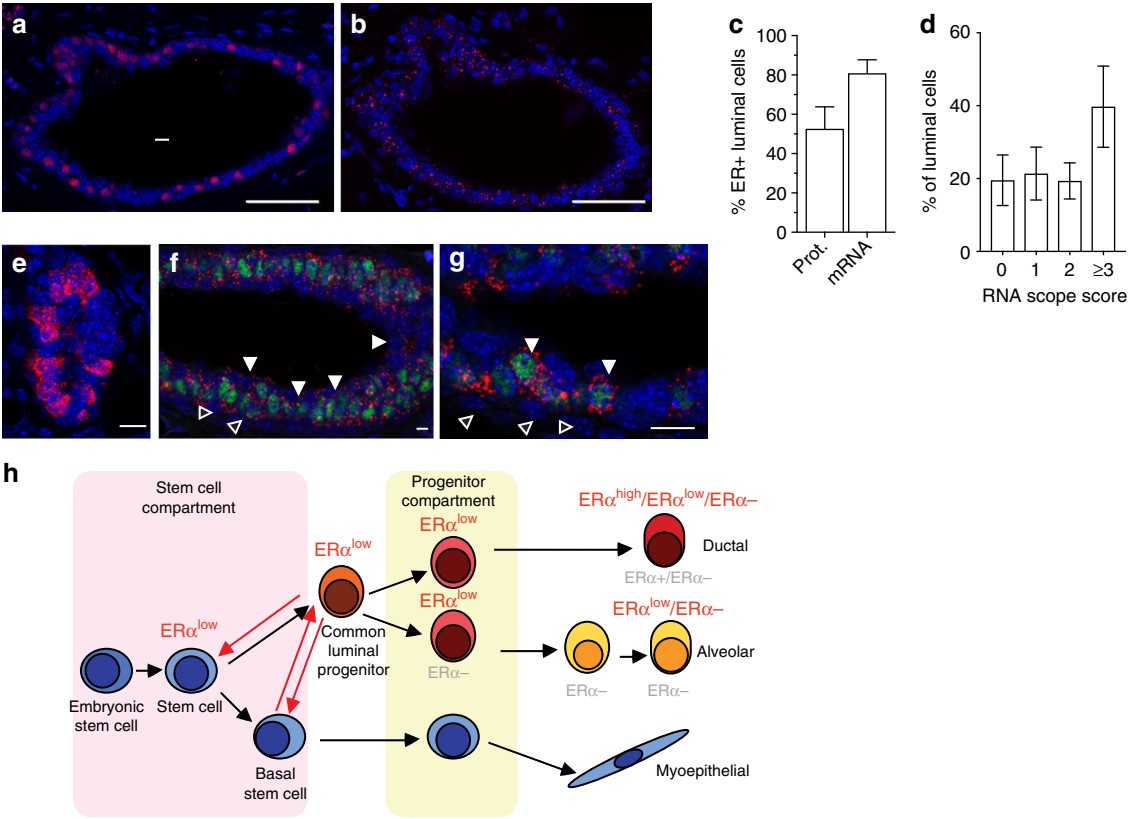

**Fig. 9** *ERα* mRNA and protein expression *ERα*$^{-/-}$ and *WT* MECs. **a** ERα immunofluorescence (red) and **b** RNAscope with anti-*Esr1* probes (red) on an adjacent section counterstained with DAPI (blue) on mammary gland from 6-week-old *WT* female. Scale bars; 50 μm. **c** Bar plot showing percentage of luminal cells with detectable levels of ERα protein and *Esr1* mRNA at puberty (*n* = 5; mean ± SD). **d** Bar plot showing percentage of luminal cells with different *Esr1* RNAscope scores (*n* = 5; mean ± SD). **e** RNAscope with anti-*Areg* probes (red) counterstained with DAPI (blue) on mammary gland from 6-week-old *WT* female. Scale bar; 50 μm. **f** RNAscope with anti-*Esr1* or **g** anti-*Areg* probes (red) co-stained with anti-PgR antibody (green) and counterstained with DAPI (blue). Arrowheads point to apically-located luminal cells with *ERα* transcript but no PgR expression (**f**) and *Areg* transcript and PgR expression (**g**). Empty arrowheads point to myoepithelial cells **f**, **g**. Scale bars; 100 μm. **h** Current model of mammary stem cells and the differentiation hierarchy after Visvader, J.E. and Stingl, J.[56]. In grey: previously proposed ERα status of different MEC differentiation stages. In red: ERα status based on the present findings

with high *Esr1* mRNA levels as determined by RNAscope amounted to 40% whereas the percentage of ERα+ cells by IHC in adjacent sections was somewhat higher with 50% suggesting that mRNA and protein levels do not directly correlate. Whether the ERα$^{high}$ and ERα$^{low}$ luminal cell populations that we propose correspond to the mature and progenitor sensor cells recently discerned by single cell RNA sequencing[39,40] remains to be tested.

Similarly, whether the different ERα states mark distinct cell populations or reflect different transient and/or functional states of otherwise similar luminal cells, will need to be addressed by more in depth single cell sequencing. Lineage tracing experiments have shown that ERα+ cells only give rise to ERα+ cells[41,42]. In one approach, the prominin promoter was used to drive Cre expression and shown to be expressed in a subset of ERα$^{high}$ cells[42]. In the other approach, a 4-kb fragment upstream of the *Esr1* transcription start site was used to drive doxycycline-inducible transgenic Cre expression; subsequently the founder was selected which showed the best overlap of Cre expression with ERα$^{high}$ cells[41]. It will be interesting to see how the outcome would change if the ERα$^{low}$ luminal cell populations were traced. If indeed would these cells give rise exclusively to ERα$^{low}$ luminal cells suggesting that ERα status is fixed or they may be less differentiated and more plastic and be able to give rise also to ERα-cells.

Furthermore, it remains to be ascertained with more sensitive quantitative approaches whether ERα status is truly tripartite in

the luminal epithelium or whether underlying our observations is a continuous, possibly changing gradient in ERα expression levels.

The current model, about the cellular origin of breast cancer has it that ERα+/luminal breast cancers originate from ERα+ while triple negative/basal-like breast cancers from ERα-cells. Our findings beg the question how the ERα$^{low}$ vs. ERα$^{high}$ luminal cells contribute to breast cancer development. Similarly, the current view of ERα as a marker of differentiation in a hierarchical model for mammary cell types appears too simple. At least when expressed at low levels, ERα may have an important role in controlling cell plasticity.

A picture emerges that links ERα function and signalling mechanisms to its expression level. ERα$^{high}$ sensor cells require both AF-1 and AF-2 to transcribe essential paracrine mediators, like *Areg*, *Wnt4* and *Pgr1* and to induce ductal growth. It is tempting to speculate that this is a means of ensuring that the strongly pro-proliferative actions of paracrine signalling only kick in when both sufficient levels of the ER ligand itself are around and growth factor signalling is simultaneously active resulting in ER phosphorylation and activation of AF-1. ERα$^{low}$ responder cells largely rely on AF-2 to transcriptionally control the cytoskeleton, cell adhesion, and signalling, essential for the expansion of this cell population during ductal morphogenesis with its extensive cell movements in response to paracrine signalling. The effects of AF-1 deletion appear more subtle in the responder cells.

We failed to detect a phenotype in the context of chimeric epithelia during ductal outgrowth, yet the *AF-1*[0] MECs were less efficient than their WT counterparts during intraductal engrafting suggesting that AF-1 may affect the expression of cell adhesion genes a hypothesis that seems plausible in light of the altered CD24 and CD49f expression in this mutant.

During pregnancy, *ERα*[−/−] MEC populations expand more than their *WT* counterparts. However, we failed to detect a difference in proliferation indices at the specific pregnancy time point we analysed, e.g., day 12.5. Two scenarios are conceivable; either at an earlier or a later time point, f.i. during alveologenesis the *ERα* MECs have higher proliferative indices than their *WT* counterparts, this can be addressed by more detailed follow up studies. Alternatively, pregnancy-induced cell proliferation may be faster in ERα negative MECs than in ERα[low] or ERα[high] cells. As only 20% of the luminal cells in the WT setting are ERα negative whereas in the *ERα*[−/−] MECs 100% lack ERα expression this could underlie the observed compensation in the course of pregnancy.

The opposite biological effects in puberty and pregnancy may also reflect a dose-dependent biphasic mode of action with low oestrogen levels in puberty and high levels during pregnancy triggering opposite actions. Alternatively, it may relate to the specific hormonal context, which changes dramatically from puberty to pregnancy, this concerns a multitude of hormones but is particularly striking for progesterone. In addition to changes in the levels of the ligands the concomitant loss of PgR expression, which opposes many ERα actions[43] may have a role to play.

Previously, *ERαNeoKO* mice[44] were suggested to be an AF-1 deficient mouse model as they transcribe a spliced mRNA that gives rise to a receptor lacking parts of the domain A and all domain B[45]. In contrast to the present finding that *ERαAF-1*[0] mammary epithelia fail to develop alveoli, some of the *ERαNeoKO* epithelial grafts grew during pregnancy and developed alveoli[16]. It is conceivable that complete loss of ERα function in some MECs accounts for this; the activity of the resulting E1 ERα variant protein in the *ERαNeoKO* mice is animal and cell-type dependent[46].

Our finding that all ERα signalling is AF-2-dependent provides a molecular basis for the breast specific efficacy of the widely used breast cancer therapeutic, tamoxifen, which is an AF-2 antagonist but AF-1 agonist[47].

## Methods

**Mice**. All mice were maintained and handled according to Swiss guidelines for animal safety with a 12-h-light-12-h-dark cycle, controlled temperature and food and water ad libitum. Animal experiments were performed in accordance with protocols approved by the Service de la Consommation et des Affaires Vétérinaires of Canton de Vaud, Switzerland. The *ERα*[−/−][15], *ERαAF-1*[0][13], *ERαAF-2*[0][14] *RAG1*[−/−][48], *C57BL/6-Tg(Act-EGFP)*[27] and *tg(CAG-DsRed\*MST)1Nagy/J*[28] were maintained in C57Bl6 background. The *NOD.Cg-Prkdcscid Il2rgtm1Wjl/SzJ* mice (*NSG*) were purchased from Jackson Laboratories. Mice were anaesthetised by intraperitoneal injection with 10 mg/kg xylazine and 90 mg/kg ketamine (Graeub).

**Transplantations and intraductal injections**. For transplantations, 1 mm³ epithelial fragments were prepared under a fluorescence stereoscope from *GFP*+ donor mice and inserted into the inguinal fat pads of 3-week-old *Rag1*[−/−] females cleared of their endogenous epithelium as described[49]. To generate chimeric epithelia, 90,000 dissociated *WT.DsRed*+ mixed with 10,000 dissociated *mutant.GFP*+ cells in 10 μL of 20% growth factor reduced matrigel (BD Biosciences) were injected into cleared fat pads. Mutant and control fragments were grafted into contralateral glands. The outgrowths were analysed 10 weeks after transplantation or after impregnation of the hosts on P16–18. Progesterone and control pellets were prepared as described[50] and inserted subcutaneously in the neck region of 12-week-old females. Intraductal injections were performed as described[32].

**Mammary gland whole-mounts and image analysis**. Mammary gland whole-mounts were performed as described[51]. Stereomicrographs were acquired using a LEICA MZ FLIII stereomicroscope with Leica MC170 HD. Fluorescence images were acquired using a LEICA M205FA fluorescence stereomicroscope with Leica DFC 340FX camera. The area of mammary fat pad filled by ducts and branching points were determined using ImageJ software. To determine the area of fat pad filled with epithelium, areas with ducts were circled using ImageJ software. The total area of the mammary fat pad, filled and not filled, was measured using the same method and the percentage of fat pad occupancy determined as ratio between duct area/total fat pad area. GFP and DsRed signal intensities were calculated with (integrated density$_{signal}$ − area$_{signal}$) × mean$_{background}$ using ImageJ software.

**Immunofluorescence and antibodies**. Glands were fixed with 4% paraformaldehyde overnight at 4 °C and embedded in paraffin. Sections measuring 4 μm were dewaxed, rehydrated and subjected to antigen retrieval with 10 mM trisodium citrate buffer, pH 6.0 for 20 min at 95 °C. Blocking of 1 h with 1% BSA was followed by incubation with primary and secondary antibodies. Primary antibodies were: rabbit anti-ERα (1:100–400; MC20, sc-542 SantaCruz), rabbit anti-PR (1:400; SP2, RM-9102 Thermo Fisher Scientific), anti-SMA (1:100; RB-9010-P0; Thermo Fisher Scientific), anti-p63 (1:100; MU418-UC; BioGenex) and rabbit anti-RFP (1:400; cat# PM005, MBL). All secondary antibodies were used at 1:500 dilutions (Molecular probes): alexa 488-conjugated anti-rabbit IgG, alexa 488-conjugated anti-goat IgG, alexa 568-conjugated anti-rabbit IgG and alexa 568-conjugated anti-mouse IgG. Nuclei were counterstained for 10 min with DAPI (Sigma) and mounted with Dabco (0718, Carl Roth). Images were acquired on confocal Zeiss LSM700 and reassembled with ImageJ software.

**RNA extraction and quantitative reverse transcription polymerase chain reaction(PCR) analysis with sequencing and bioinformatics analysis**. Mammary glands #3–5 were homogenised in TRIzol (15596026, Invitrogen), total RNA was isolated using miRNeasy Micro Kit (217084, Qiagen), and cDNA synthesised with 250 ng of total RNA using SuperScript VILO cDNA synthesis kit (11754-050 Invitrogen). For high-throughput qPCR, after a step of pre-amplification PCR using TaqMan PreAmp Master Mix kit (PN 4384556A Applied Biosystems), semi-quantitative real-time PCR analysis in duplicates was performed using EvaGreen DNA-binding dye with 48.48 dynamic arrays (Fluidigm) on Biomark HD machine (Fluidigm). Primers used for the pre-amplification are composed of a mix of all primers used at a final concentration of 500 nM. Data were analysed and normalised to three housekeeping genes (*Gapdh, 36B4* and *Hprt*) using GenEx software (MultiD). Primer sequences: *Areg*, ACC AAT GAG AAC TCC GCT GCT, AAG CGA TTC GCC TTT CCC TGA, *36B4*, GAA CTT GCT GCA TAG CAG ACC, CTC CTT GCA ATC TCC CAG AG, *Hprt*, ACG AGA GGC TCA CTG CAG AC, GGA GAT TGC GGG TTT ATA ATG, *Wnt-4*, AGG AGT GCC AAT ACC AGT TCC, CAG TTC TCC ACT GCT GCA TG, *Prlr*, GAT CAT TGT GGC CGT TCT CT, CCA GCA AGT CCT CAC AGT CA, *Pgr*, AAA CTG CCC AGC ATG TCG TCT, GCT CTC GTT AGG AAG GCC CA, *Itgb1*, TTC AGA CTT CCG CAT TGG CTT TGG, TGG GCT GGT GCA GTT TTG TTC AC, *Gapdh*, CCA ATG TGT CCG TCG TGG ATC, GTT GAA GTC GCA GGA GAC AAC.

RNA was extracted from GFP + FACS-sorted cells using miRNeasy Mini Kit (Qiagen). Libraries were prepared in two steps. The first step was performed with SMART-Seq v4 ultra low input RNA Kit (Clontech). Briefly, 10 ng of RNA was reverse transcribed using a oligo dT primer flanked with a proprietary adaptor. Template switching mechanism was then used to append another proprietary adaptor on the 3′ end of the cDNA corresponding to the beginning of the mRNA molecule. A PCR specific for the aforementioned adaptors was used to create and amplify double-stranded cDNA molecules. The second step was performed with Nextera XT kit (Illumina). Briefly, tagmentation of the double-stranded cDNA with hyperactive Tn5 created fragments of a few hundred bp, flanked with Illumina proprietary adaptors. DNA was then PCR amplified with Illumina primers for eight cycles, generating final libraries of ~400 bp (insert plus adaptors). Libraries were sequenced on Illumina NextSeq 500 instrument with single-end reads of 85nt. Base calls and Illumina adaptors trimming performed using bcl2fastq v2.18. Clontech adaptors trimming performed with CLC 9. RNAseq reads were aligned to the mm10 genome assembly using the web application HTSstation[52]. For the differential RNAseq expression analysis we applied the automated analysis pipeline ASAP[53] for 90% top expressed threshold (% of genes kept in the data frame). Lists of DE analyses were performed using edgeR package. Heatmap was generated using ClustVist[54].

**RNA in situ hybridisation**. RNAscope assay (Advanced Cell Diagnostics, Cat. No. 323110) was performed according to manufacturer's protocol on 4 μm deparaffinized sections and hybridised with probes: *Mm-Esr1* (ACD, Cat. No. 432861), Mm-Areg (ACD, Cat. No. 430501), Mm-Ppib (ACD, Cat. No. 313911, positive control) and DapB (ACD, Cat. No. 310043, negative control) at 40 °C for 2h and revealed with TSA Plus-Cy3 (Perkin Elmer, Cat. No. NEL744001KT). Rabbit anti-PgR (1:400, clone SP2, Thermo Fisher, Cat. No.: RM-9102-P) and rabbit anti-ERα (1:100; MC20, sc-542 SantaCruz) was incubated overnight at 4 °C and detected with Alexa 488 or Alexa 568 conjugated goat anti-rabbit (1:1000, Life Technology), respectively. Images were captured on confocal Zeiss LSM700 and spots quantified using QuPath and an in-house script, code available from O. Burri at the Bioimaging and Optics platform, EPFL, based on the guide for RNAscope Data Analysis.

**Hormone measurements**. Testosterone, androstenedione, 17α-hydro-xyprogesterone, corticosterone, 11-deoxycorticosterone and progesterone levels were measured by LC–MS High Resolution (Q-Exactive, ThermoFisher Scientific). Frozen plasma samples were thawed, vortex mixed, centrifuged at 33,000$g$ for 5 min. An aliquot of 50–100 μL was spiked with 10 μL of internal standards. After diluting the samples with 5% (w/v) phosphoric acid, the analytes were purified using a solid phase extraction (Oasis MCX 96-well plate, Waters). The washing steps were 5% (w/v) NH$_4$OH and 20% (v/v) methanol. The separation column was an Acquity UPLC HSS T3 column (1.8 μm, 1.0 × 100 mm, Waters), the mobile phase comprised H$_2$O and 0.01% formic acid in methanol. The liquid chromatography system was coupled to a Q-Exactive Orbitrap mass spectrometer (Thermo Scientific) using a full scan acquisition. The calibrants were certified standards (ChromSystems). Estradiol was extracted from 100 to 200 μL plasma with ethyl acetate: hexane (3:2). The upper organic layer was evaporated under nitrogen followed by derivatization with dansyl chloride. The separative column was a Zorbax Eclipse Plus RRHD C18 (Agilent); analysis was carried out on a 6495 Triple Quad LC/MS–MS (Agilent)[55].

**Fluorescence activated cell sorting**. Mammary glands were pooled for the preparation of single cell suspensions and processed for flow cytometry as described[34]. The following conjugated antibodies were used: anti-CD24-PE-Cy7 (560536, BD Pharmingen), anti-CD49f-APC (313616, Biolegend), anti-CD31-BV421 (563356, BD Pharmingen), anti-CD45-BV421 (563890, BD Pharmingen) and anti-Ter119-BV421 (563998, BD Pharmingen). Mammary epithelial cells were sorted on a FACSAria flow cytometer (Becton Dickinson) or analysed on LSRII flow cytometer analyser (Becton Dickinson).

**Statistical analysis**. Statistical analyses were performed with Prism7 software (GraphPad). Data are shown as means ± SD, or as otherwise specified. Statistical significance is indicated as follows $^*p < 0.05$, $^{**}p < 0.01$, $^{***}p < 0.001$, $^{****}p < 0.0001$, n.s. not significant.

## Data availability
The transcriptomics data have been deposited in the Gene Expression Omnibus database under accession code GSE103664;.

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

## Acknowledgements

We thank J. Dessimoz and O. Burri for advice on RNAscope and the EPFL core facilities for technical assistance, N. Hynes, F. Lenfant and Brisken laboratory members for reading of the manuscript, R. Jeitziner for help with the bioinformatic analysis. S.C., D.A. and V.S. received funding from the Swiss Cancer Ligue KFS-3701-08-2015 and SNF 31003A_162550/1 Hormonal and cell signaling control of mammary gland morphogenesis: The role of Adamts18 in epithelial-basal membrane interactions that control the stem cell function downstream of progesterone receptor signaling, G.S. and P.A. by Biltema and ISREC Foundation.

## Author contributions

Investigation: S.C., D.A., G.S., S.S., P.A., A.A. and V.S. Bioinformatic analysis: G.S. Resources: H.H., A.K., P.C. and M.F. Writing: S.C. and C.B. Funding acquisition: C.B.

## Additional information

**Competing interests:** The authors declare no competing interests.

