## [Peer Review File · Nature Communications]

Reviewers' Comments:

Reviewer #1:

Remarks to the Author:

In the manuscript by Cagnet et al. titled "The Estrogen Receptor α and its AF-1 and AF-2 domains have cell population-specific functions in the mammary epithelium" the authors use various mouse models with deficient Estrogen receptor alpha (ER α) function or expression to study the role of ER α in mammary epithelial cell development. The authors argue that they have identified a new cell population which expresses the ER α mRNA at low levels but not the protein which is required for ductal growth and controlling the alveolar proliferation in response to pregnancy. Unfortunately, I do not think they present sufficient data to support these claims. In general, the manuscript is not clear and does not give sufficient detail to support the claims. Therefore, I do not believe the study is suitable for publication at Nature Communications in its current format. I include below some suggestions which I hope the authors will find helpful.

- 1) The introduction and justification of the experimental setups used need to be substantially improved. It is not clear from the current text why this study is important and why this approach addresses the question.
- 2) Throughout the paper the authors use image analysis to quantify % occupancy of ducts/epithelial per fatpad. I think more information is needed on how exactly that was done. The information provided in the methods section is not enough. It is not clear how accurate are these measurements.
- 3) As a follow on from that point the pie charts in figure 3 and 4 rely heavily on that measurement however, if you look at the data in Figure 3i,j it seems that the numbers don't match.
- 4) How were the cells prepared for the intraductal and chimeric injections? Were they FACS sorted? If yes, on what markers? Need to show FACS plots. Also, how did you get that many cells if AF1 and AF2 dramatically affect the epithelial outgrowth.
- 5) The RNAscope data in Figure 3 needs to be quantified and distribution of staining intensity needs to be plotted. The authors provide no experimental evidence to support the claim of functional responder cells. There are many reasons why cells express RNA but not protein.
- 6) The authors also claim that AF2 domain is pro-proliferative during puberty and anti-proliferative during pregnancy. Again they need evidence for these claims they rely on image analysis in Figure 4 but FACS in figure 3. I think this needs to be addressed experimentally with BrdU/EdU labelling.
- 7) The data in Figure 5 is strong. I would have liked to see AF1 and AF2 cells also injected. This experiment might help support the claims made in Figure 3 and 4.

Reviewer #2:

Remarks to the Author:

The manuscript by Cagnet et al is an informative work in understanding the role of the estrogen receptor (ER) in the mammary gland during postnatal development and during pregnancy. This paper dissects the role of two important domains of the ER (AF1 and AF2) and how each of them is involved. Overall, the study is well described, creative and useful to the field. The authors' conclusion that ER has a biphasic response that stimulates growth during puberty and inhibits growth during pregnancy is well supported by the data. They also show that there are 3 different types of luminal cells that have different levels of ER expression: High, low, and ER0, and that in ER high and low the involvement and role of the AF1 and AF2 domains are distinct. It is interesting that overall ER expression levels are comparable among the WT, AF1 and AF2 mutants (Fig 1d).

Although the same promoters are involved, one might expect protein stability to be different, but this observation does agree with published uterine IHC. The AF1 and AF2 domains are both important in ER high cells, whereas ER low cells are AF2 domain dependent and AF1 independent. While the figures convincingly illustrate this last point, a more descriptive and detailed explanation of the role of AF1 would help the reader better understand its role in ER high cells. Also, the observed IHC high/low distinction is somewhat arbitrary and likely to be antibody dependent. It seems fortuitous that cells are neatly grouped into high vs low. One might expect a range of ERalpha expression. Also, the use of PgR as a surrogate for ERalpha activity also assumes a 1:1 correlation with a certain level of ER expression.

The hypothesis that ER function in low ER cells is biphasic is very intriguing. It would be useful to know more about the differential molecular mechanisms that are responsible.

Figure 3 is a bit confusing because some of the sub-figures are misidentified (3j) or missing (3k). In the stereo micrograph of the chimeric glands in figure 3f, it appears that the GFP signal from the AF20-GFP/WT-red chimera is weak compared to WT-GFP/WT-red and AF10-GFP/WT-red. Figure 3g shows a quantification of the GFP signal over the red signal. It might be more useful to show a dot plot of percent area of GFP signal over red signal, and might show statistical significance between WT-GFP/WT-red and AF20-GFP/WT-red, which has a p value of 0.55.

Page 6, line 2 mentions figure 3k, which is not present in figure 3.

Also, there is no figure 6k.

Reviewer #3:

Remarks to the Author:

- The authors put forth an elegant study to dissect the role of AF-1 versus AF-2 signalling in mammary gland development. The experiments performed are predominantly descriptive in nature, utilising innovative GEM models, mixing experiments and intraductal transplantation techniques to tease apart the roles of ERa, AF1 and AF2 in mammary development and pregnancy.

Major points

The co-mixing experiments do not take into account the different proportions of Mammary stem cells in each of the GEM models studied. The MaSC are the MEC subtype that are responsible for the MEC outgrowth. Differing proportions of MaSCs may potentially account for the differences seen in epithelial regeneration and fat pad filling rather than solely the genetic background of the MECs. The authors measured the basal and luminal populations of the chimeric glands, it would be useful to know the proportions of these of each GEM model prior to the mixing experiments.

What was the distribution of MaSC, luminal progenitor and mature luminal cell subpopulations? Stingl et al Nature 2006, Shackleton et al Nature 2006.

What is the role of the large differences in circulating hormonal levels in comparative studies between pregnancy and before pregnancy? Could this account for the different phenotypes seen in different MECs between puberty and pregnancy?

The discussion on the impact of their findings relating to the mammary epithelial hierarchy could be further supported by an analysis of the three mammary epithelial subsets as initially put forward by Stingl and Shackleton, as well as consideration for the limiting dilution mammary fat pad transplantation assays. In this study, only the basal and luminal subsets were studied in the chimeric models.

While there was a discussion about the need for a revision of the current model for the origins of

various subtypes of breast cancer, this was speculative as the study did not compare cancer models to their findings.

Minor points

In the methods section, it states that progesterone pellets were used. In which experiments did this occur? What dose were these pellets?

Page 5, Line 2. Fig 3C, D is mislabelled as Fig 3K

Reviewer 1:

We thank the reviewer for his/her constructive comments and suggestions, which we addressed as detailed below:

In the manuscript by Cagnet et al. titled "The Estrogen Receptor α and its AF-1 and AF-2 domains have cell population-specific functions in the mammary epithelium" the authors use various mouse models with deficient Estrogen receptor alpha (ER α) function or expression to study the role of ER α in mammary epithelial cell development. The authors argue that they have identified a new cell population which expresses the ER α mRNA at low levels but not the protein which is required for ductal growth and controlling the alveolar proliferation in response to pregnancy. Unfortunately, I do not think they present sufficient data to support these claims. In general, the manuscript is not clear and does not give sufficient detail to support the claims. Therefore, I do not believe the study is suitable for publication at Nature Communications in its current format. I include below some suggestions which I hope the authors will find helpful.

1) The introduction and justification of the experimental setups used need to be substantially improved. It is not clear from the current text why this study is important and why this approach addresses the question.

Major comments:

1. We have substantially rewritten the manuscript to better explain the rationale and importance of the study. We have provided additional information to justify and explain the experimental set-ups. As examples, we provided additional explanations for the choice of experimental time points on p 4 and integrated the analysis of the hormone measurements into the main text p5 to illustrate the importance of the transplantation approach to discern the epithelial-intrinsic role of the ER mutants.

We have reorganized the structure of the manuscript to improve on its clarity.

2) Throughout the paper the authors use image analysis to quantify % occupancy of ducts/epithelial per fatpad. I think more information is needed on how exactly that was done. The information provided in the methods section is not enough. It is not clear how accurate are these measurements.

2. We now state in the text (page 7, last paragraph) that the occupancy measurements, which were used to summarize the analysis of the chimera experiments are only approximative. Color contributions to the 3D mammary glands were quantified based on a single 2D image. We removed the statistical analysis (p values) from these data (Figures 3d,e, 4c,d, 5f) not to mislead the reader and clarify that this is just an approximative rendering of the experimental outcome as analysed by fluorescence stereomicroscopy. Importantly, we now show quantitative data for all 4 genotypes in nulliparous and pregnant hosts (Figure 5i, j) obtained by FACS analysis. More information about the image analysis is found in the method sections p15, 1st paragraph.

3) As a follow on from that point the pie charts in figure 3 and 4 rely heavily on that measurement however, if you look at the data in Figure 3i,j it seems that the numbers don't match.

3. The reviewer pointed out that the results for basal and luminal cells in Figure 3f,g do not match. Importantly, the quantification of luminal cells (Figure 3f), which account for most of the fluorescent signal detected by stereoscopic imaging is consistent with the approximate results shown in Figure 3d,e. The fluorescence emanating from myoepithelial cells is hardly appreciated by fluorescence stereomicroscopy.

4) How were the cells prepared for the intraductal and chimeric injections? Were they FACS sorted? If yes, on what markers? Need to show FACS plots. Also, how did you get that many cells if AF1 and AF2 dramatically affect the epithelial outgrowth.

4. For the intraductal injections and for generating chimeric epithelia, donor mammary glands were mechanically and enzymatically dissociated. During subsequent centrifugation steps, the floating adipocytes were removed together with the fibroblasts in the supernatant. The mixture of cells was processed to single cells but not further purified by FACS sorting or other.

We now explain in the text that the low cell numbers obtained from the ER mutant prevented us from FACS purifying subpopulations (page 10, lines 17 ff). Importantly, we have now analyzed mammary glands from all 3 ER mutants and respective *WT* littermates by FACS and show that the mutations severely affect CD24 and CD49f expression (Figure 5m). As a result the typical FACS profile is distorted.

From the ER mutant mice we obtained only about 10% of the cell numbers obtained from *WT* littermates. However, as the mixing ratio was 1:10 this was not a problem.

5) The RNAscope data in Figure 3 needs to be quantified and distribution of staining intensity needs to be plotted. The authors provide no experimental evidence to support the claim of functional responder cells. There are many reasons why cells express RNA but not protein.

5. We have quantified the RNAscope data (Figure 7c) and show the RNAscope score (Figure 7d) and also added ER IHC on adjacent sections (Figure 7a) which we quantified (Figure 7c).

We have added additional explanation and provide two additional references 24, 25 on page 7, 2nd paragraph to highlight the paracrine mechanism of actions based on which the concept of sensor and responder cells is based.

We concur with the reviewer about there being many reasons why cells express RNA but not protein. We propose that ER protein may be highly turned over in the cells which show mRNA expression but are negative by IHC. It is also conceivable that the ER is present also in other cellular compartments at levels that are below the threshold of IHC, as has been suggested for the membrane-bound ER fraction or it is in a state in which the epitopes recognized by the IHC antibody are masked.

6) The authors also claim that AF2 domain is pro-proliferative during puberty and anti-proliferative during pregnancy. Again they need evidence for these claims they rely on image analysis in Figure 4 but FACS in figure 3. I think this need to be addressed experimentally with BrdU/EdU labelling.

6. The reviewer raised the very important point that our claim that AF2 domain is pro-proliferative during puberty and anti-proliferative during pregnancy needs to be sustained experimentally with BrdU/EdU labelling. We have analysed cell proliferation in nulliparous and pregnant hosts by Ki67 and Edu labelling of additional samples (Figure 5k,l). It shows that there is no significant difference in cell proliferation between *WT* and *ER*^{-/-} mammary epithelial cells. This has made us revisit the interpretation of the biphasic effects of ER and we have corrected the use of the terms pro-and anti- proliferative throughout the text.

7) The data in Figure 5 is strong. I would have liked to see AF1 and AF2 cells also injected. This experiment might help support the claims made in Figure 3 and 4.

7. We have performed additional in vivo experiments with AF-1⁰ and AF-2⁰ mammary epithelial cells and now provide quantitative data on all 3 different ER genotypes both in nulliparous and pregnant hosts (Figure 5i, j).

Reviewer 2:

The manuscript by Cagnet et al is an informative work in understanding the role of the estrogen receptor (ER) in the mammary gland during postnatal development and during pregnancy. This paper dissects the role of two important domains of the ER (AF1 and AF2) and how each of them is involved. Overall, the study is well described, creative and useful to the field. The authors conclusion that ER has a biphasic response that stimulates growth during puberty and inhibits growth during pregnancy is well supported by the data. They also show that there are 3 different types of luminal cells that have different levels of ER expression: High, low, and ER0, and that in ER high and low the involvement and role of the AF1 and AF2 domains are distinct. It is interesting that overall ER expression levels are comparable among the WT, AF1 and AF@ mutants (Fig 1d). Although the same promoters are involved, one might expect protein stability to be different, but this observation does agree with published uterine IHC. The AF1 and AF2 domains are both important in ER high cells, whereas ER low cells are AF2 domain dependent and AF1 independent. While the figures convincingly illustrate this last point, a more descriptive and detailed explanation of the role of AF1 would help the reader better understand its role in ER high cells. Also, the observed IHC high/low distinction is somewhat arbitrary and likely to be antibody dependent. It seems fortuitous that cells are neatly grouped into high vs low. One might expect a range of ERalpha expression. Also, the use of PgR as a surrogate for ERalpha activity also assumes a 1:1 correlation with a certain level of ER expression.

The hypothesis that ER function in low ER cells is biphasic is very intriguing. It would be useful to know more about the differential molecular mechanisms that are responsible.

Figure 3 is a bit confusing because some of the sub-figures are misidentified (3j) or missing

(3k). In the stereo micrograph of the chimeric glands in figure 3f, it appears that the GFP signal from the AF20-GFP/WT-red chimera is weak compared to WT-GFP/WT-red and AF10-GFP/WT-red. Figure 3g shows a quantification of the GFP signal over the red signal. It might be more useful to show a dot plot of percent area of GFP signal over red signal, and might show statistical significance between WT-GFP/WT-red and AF20-GFP/WT-red, which has a p value of 0.55.

Page 6, line 2 mentions figure 3k, which is not present in figure 3.

Also, there is no figure 6k.

We thank the reviewer for the appreciation of our work as well described, creative and useful and his/her constructive comments and suggestions.

Regarding the role of AF-1 we have added some explanations on page 8, second paragraph and in the discussion page 13, lines 9-12, 15-20.

We agree with the reviewer that the IHC high/low distinction is arbitrary and likely antibody dependent. Indeed, we observe in IHC and IF that the intensity of the staining varies among individual mammary epithelial cells. Hence, it is conceivable that there is a gradient of ER expression, as we now conclude, page 12 line 10 and last line of the same page.

Furthermore, we have addressed the concern that the PgR is not a perfect surrogate for ER expression and used sections adjacent to those used for RNAscope to perform ER IF (figure 7a). Quantification of both now shown in figure 7c demonstrates that 50% of the luminal cells are ER+ by IF and 80% are ER+ by RNAscope.

We now discuss more extensively the different mechanisms that may account for the biphasic effect of the ER pages 14/14.

We thank the reviewer for the helpful suggestion to improve on the image analysis. In fact, we have altogether removed the statistical analysis (p values) from these data (Figures 3d,e, 4c,d, 5f) because the data derived from the stereomicrographs are approximative as the quantifications were based on 2D images of the 3D objects. We now provide quantitative data of the contribution of mutant and *WT* cells that are based on FACS analyses in Figure 5i,j.

The labeling of Fig. 3 its legends and the references in the text have been corrected.

We have corrected the wrong labeling Fig. 3k to Fig. 3c.

We have corrected the wrong labeling Fig. 6k.

Reviewer 3:

- The authors put forth an elegant study to dissect the role of AF-1 versus AF-2 signalling in mammary gland development. The experiments performed are predominantly descriptive in

nature, utilising innovative GEM models, mixing experiments and intraductal transplantation techniques to tease apart the roles of ER α , AF1 and AF2 in mammary development and pregnancy.

Major points

The co-mixing experiments do not take into account the different proportions of Mammary stem cells in each of the GEM models studied. The MaSC are the MEC subtype that are responsible for the MEC outgrowth. Differing proportions of MaSCs may potentially account for the differences seen in epithelial regeneration and fat pad filling rather than solely the genetic background of the MECs. The authors measured the basal and luminal populations of the chimeric glands, it would be useful to know the proportions of these of each GEM model prior to the mixing experiments.

What was the distribution of MaSC, luminal progenitor and mature luminal cell subpopulations? Stingl et al Nature 2006, Shackleton et al Nature 2006.

What is the role of the large differences in circulating hormonal levels in comparative studies between pregnancy and before pregnancy? Could this account for the different phenotypes seen in different MECs between puberty and pregnancy?

The discussion on the impact of their findings relating to the mammary epithelial hierarchy could be further supported by an analysis of the three mammary epithelial subsets as initially put forward by Stingl and Shackleton, as well as consideration for the limiting dilution mammary fat pad transplantation assays. In this study, only the basal and luminal subsets were studied in the chimeric models.

While there was a discussion about the need for a revision of the current model for the origins of various subtypes of breast cancer, this was speculative as the study did not compare cancer models to their findings.

We thank the reviewer for the appreciation of our work as elegant study and his/her constructive comments and suggestions.

Major comments:

Following this reviewer's suggestion to analyze the distribution of MaSC, luminal progenitor and mature luminal cell subpopulations as done by Stingl et al Nature 2006, Shackleton et al Nature 2006, we have subjected mammary glands from *WT*, *AF-1⁰*, *AF-2⁰*, and *ER^{-/-}* mice to FACS analysis. As shown (Figure 5m) ER mutations results in differences in CD24 and Cd49f expression, resulting in a major shift of the populations, which makes it impossible to align the FACS profiles of the ER mutant cells with those of the *WT* cells.

However, in line with this reviewer's suggestion that a stem cell phenotype may account for the inefficiency of *ER^{-/-}* MECs to contribute to ductal outgrowth, we now show that the proliferative indices do not differ between *ER^{-/-}* and *WT* MECs (Figure 5k) and discuss that indeed a decreased stem cell number and/or impaired stem cell function

may account for the decreased ability of ER and AF-2 cells to contribute to ductal outgrowth p10 lines 8ff.

Indeed, as the reviewer suggests, differences in circulating hormone levels between nulliparous and pregnant mice may account for the different phenotypes. This is discussed p14 lines 3-5.

As suggested by the reviewer, we have analysed the mammary epithelial subsets put forward by Stingl and Shackleton and find that all 3 ER mutants strongly affect the expression of CD24 and CD49f resulting in major shifts in the populations (Figure 5m). This precluded any conclusion about specific cell subpopulations.

We have rephrased the sentence about the origins of various subtypes of breast cancer. Instead of saying the model may need revision when we did not compare cancer models we merely raise the question what the role of ER high versus ER low cells may be in disease development page 13 lines 1-4.

Minor points

In the methods section, it states that progesterone pellets were used. In which experiments did this occur? What dose were these pellets?

Page 5, Line 2. Fig 3C, D is mislabelled as Fig 3K

Minor comments:

The progesterone pellets contained 20 mg of progesterone and were used for the experiment shown in Supplemental Fig. 3. We came to realized that we had omitted the text referring to Supplemental Fig. 3 and have added it now p6 last paragraph/p7 top.

The labeling of Fig. 3 its legends and the references in the text have been corrected.

Reviewers' Comments:

Reviewer #1:

Remarks to the Author:

I have now read the revised manuscript and indeed it is much improved. The authors have addressed the major questions raised.

I have one further point - I think the authors need to add few sentences in the discussion reconciling their data with the recent ER α lineage tracing study from the Blanpain lab (Van Keymeulen A. et al 2017 Cell Reports). I think this discussion will be useful for the field.

Reviewer #2:

Remarks to the Author:

In this revised manuscript, the authors have adequately addressed the comments and concerns of this reviewer. The study addresses and provides convincing data to describe the biphasic role of the estrogen receptor (ER), especially the differential roles of AF1 and AF2 in the mammary gland during postnatal development and during pregnancy.

Reviewer #3:

None